# Depletion of Endothelial-Derived 2-AG Reduces Blood-Endothelial Barrier Integrity via Alteration of VE-Cadherin and the Phospho-Proteome

**DOI:** 10.3390/ijms25010531

**Published:** 2023-12-30

**Authors:** Aidan A. Levine, Erika Liktor-Busa, Shreya Balasubramanian, Seph M. Palomino, Anya M. Burtman, Sarah A. Couture, Austin A. Lipinski, Paul R. Langlais, Tally M. Largent-Milnes

**Affiliations:** 1Department of Pharmacology, College of Medicine, University of Arizona, Tucson, AZ 85724, USA; aidanlevine@arizona.edu (A.A.L.); erikal@arizona.edu (E.L.-B.); shreyabala31@arizona.edu (S.B.); sephmp2@arizona.edu (S.M.P.); annaburtman@arizona.edu (A.M.B.); sarahacouture@arizona.edu (S.A.C.); 2Division of Endocrinology, Department of Medicine, College of Medicine, University of Arizona, Tucson, AZ 85724, USA; lipinski1@arizona.edu (A.A.L.); langlais@arizona.edu (P.R.L.)

**Keywords:** endocannabinoid, blood–brain barrier, 2-AG, DAGLα, VE-cadherin

## Abstract

Mounting evidence supports the role of the endocannabinoid system in neurophysiology, including blood–brain barrier (BBB) function. Recent work has demonstrated that activation of endocannabinoid receptors can mitigate insults to the BBB during neurological disorders like traumatic brain injury, cortical spreading depression, and stroke. As alterations to the BBB are associated with worsening clinical outcomes in these conditions, studies herein sought to examine the impact of endocannabinoid depletion on BBB integrity. Barrier integrity was investigated in vitro via bEnd.3 cell monolayers to assess endocannabinoid synthesis, barrier function, calcium influx, junctional protein expression, and proteome-wide changes. Inhibition of 2-AG synthesis using DAGLα inhibition and siRNA inhibition of DAGLα led to loss of barrier integrity via altered expression of VE-cadherin, which could be partially rescued by exogenous application of 2-AG. Moreover, the deleterious effects of DAGLα inhibition on BBB integrity showed both calcium and PKC (protein kinase C)-dependency. These data indicate that disruption of 2-AG homeostasis in brain endothelial cells, in the absence of insult, is sufficient to disrupt BBB integrity thus supporting the role of the endocannabinoid system in neurovascular disorders.

## 1. Introduction

Alterations to the integrity of the blood–brain barrier (BBB) are well documented within neurodegenerative and neuroinflammatory disorders, such as Alzheimer’s disease and migraine headaches [1,2]. BBB integrity is maintained by the neurovascular unit (NVU), which comprises endothelial cell tight junctions, astrocyte foot processes, and pericytes [3,4]. Loss of integrity of the barrier may stem from alterations to any of these elements of NVU [5,6,7]. Increased BBB permeability has been linked to the accumulation of toxic metabolites, alterations to blood flow, cerebral edema, and other adverse outcomes; however, the processes that drive these changes are not fully understood [2,8,9].

One mechanism implicated in BBB integrity is the endocannabinoid system (ECBS). Multiple GPCRs (e.g., CB_1_R and CB_2_R), lipid metabolizing enzymes (e.g., monoacylglycerol lipase (MAGL); α/β serine hydrolase 6, (ABHD6)), and two main lipid mediators, AEA and 2-AG compose the ECBS [10,11,12]. 2-AG and AEA are produced on demand to serve as retrograde neuromodulators which act at CB_1_R and CB_2_R to negatively regulate cell-cell communication; 2-AG binds to and activates both receptors [10,12]. Synthesis of 2-AG requires activation of PLC to produce inositol triphosphate (IP_3_) and diacylglycerol (DAG) from PIP_2_ [10,12], which is followed by activation of DAGLα and β that converts DAG into 2-AG and a free fatty acid; DAGLα is the dominant isoform in the CNS [12]. In brain homogenates, 2-AG hydrolysis is mediated by MAGL (80%) and ABDH6 (6–8%) [10], which generate arachidonic acid, a precursor for inflammatory eicosanoids (e.g., PGE_2_). The ECBS serves several physiologic functions, including neuroprotection and restoration of tissue homeostasis following stress, inflammation, and tissue injury [13,14,15,16,17]. At the BBB, however, the role of the ECBS remains elusive. Activation of the cannabinoid receptors by exogenous ligands and by inhibition of 2-AG synthesis has been shown to mitigate BBB impairment in models of TBI and stroke, indicating a homeostatic role of 2-AG in BBB function [18,19,20,21]. Despite these advances in knowledge, the mechanisms by which the ECBS affects BBB structural integrity have yet to be elucidated.

To fill this gap in knowledge, we utilized an in vitro model of bEnd.3 cell monolayers to examine the effects of 2-AG depletion, via the DAGLα inhibitor LEI-106 and siRNA inhibition of DAGLα, on the synthesis of endocannabinoid lipids, monolayer functional integrity, cell viability and morphology, junctional protein expression, and changes in the phospho-proteome. We hypothesized that endothelial-derived 2-AG would maintain BBB homeostasis and that DAGLα inhibition would impair BBB integrity. Our data supports a role for endothelial-derived 2-AG in BBB endothelial cell homeostasis through the regulation of cell structure and VE-cadherin function. The impact of 2-AG depletion on kinase activity, along with cytoskeletal organization, was also supported by phospho-proteomics data. 

## 2. Results

### 2.1. Blockade of DAGLα Decreases 2-AG Levels in Endothelial Cells without Altering AEA Levels

The first experiment was designed to determine whether cultured brain endothelial cells (bEnd.3) synthesized endocannabinoid lipids, 2-AG, and AEA using LC-MS. Basal levels of 2-AG and AEA synthesized by bEnd.3 cells were determined to be 1.941 ± 0.189 nmol/g and 5.422 ± 0.371 pmol/g, respectively. We next examined whether blockade of DAGLα in the brain endothelial cells (bEnd.3) altered endocannabinoid lipid levels. Cells were treated with LEI-106 (1.3 mM equimolar to the 40 mg/kg dose in vivo) for 15 min, followed by a wash step before harvest by cell scraping (Figure 1A). The cell pellet was lysed and subjected to LC-MS to measure 2-AG and AEA levels. The treatment of bEnD.3 cells with LEI-106 significantly reduced the level of 2-AG without changing to levels of AEA compared to vehicle (Figure 1B,C; LEI-106 vs. vehicle, 2-AG: *p* = 0.001, as assessed by unpaired *t*-test, *n* = 4–10 in each group; AEA: *p* = 0.1984, as assessed by unpaired *t*-test, *n* = 4–10 in each group), suggesting on-target actions of LEI-106 at DAGLα. It is noteworthy that 2-AG and AEA were detected in the nmol and pmol ranges, respectively, mirroring tissue observations of a roughly 1000-fold difference in concentration.

To determine if LEI-106-induced depletion in 2-AG was coupled to changes in DAG levels, quantitative ELISA was performed using the same experimental setting. No significant difference was observed between LEI-106 and vehicle-treated cells (Figure 1D) (DAG level in LEI-106 (650 µM) group: 4.713 ± 0.2946 ng/mL, DAG level in LEI-106 (1.3 mM) group: 4.701 ± 0.327 ng/mL, DAG level in vehicle group: 4.259 ± 0.485 ng/mL, LEI-106 (650 µM) vs. vehicle: *p* = 0.4364; LEI-106 (1.3 mM) vs. vehicle: *p* = 0.4626, as assessed by unpaired *t*-test, *n* = 8/group).

The next investigation employed genetic manipulation of DAGLα to assess endocannabinoid levels in bEnd.3 cells to complement pharmacology with LEI-106. bEnD.3 cells were transiently transfected with siRNA targeting DAGLα or a scrambled, non-targeting control. Cells were lysed 72 h post-transfection, and lysates subjected to Western blot to detect DAGLα expression. The transfection of DAGLα-targeting siRNA decreased the detection of DAGLα protein significantly compared to the non-targeting control (Figure 1E) (DAGLα siRNA vs. non-targeting control siRNA: *p* < 0.0001, as assessed by unpaired *t*-test, *n* = 8/group). siRNA-transfected cells were assessed as well for levels of 2-AG (Figure 1F) and AEA (Figure 1G) via LC-MS. siRNA knockdown of DAGLα in bEnD.3 cells caused a significant reduction in 2-AG levels compared to non-targeting control (Figure 1F) (DAGLα siRNA vs. non-targeting control siRNA: *p* = 0.0266, as assessed by unpaired *t*-test, *n* = 4/group). However, the AEA level in bEnd.3 cells was significantly increased after DAGLα siRNA transfection compared to non-targeting control, which suggests that the cells can compensate for the loss of 2-AG by increasing AEA level (Figure 1G) (DAGLα siRNA vs. non-targeting control siRNA: *p* = 0.031, as assessed by unpaired *t*-test, *n* = 4/group). Overall, these data support that brain endothelial cells have DAGLα, make detectable 2-AG under physiological conditions, and are sensitive to treatment with LEI-106 in vitro.

### 2.2. 2-AG Depletion Decreases Trans-Endothelial Electrical Resistance

Having established that bEnd.3 cells are responsive to LEI-106, in vitro analysis of barrier integrity was performed via testing of TEER (Figure 2) and ^14^C-sucrose transport (Figure 3). For TEER experiments, bEnd.3 cells were grown to confluency on trans-well membranes and then treated with LEI-106 (15 min, 650 µM, 1.3 mM, luminal side) or KCl as a positive control (100 mM, 5 min, abluminal side) (Figure 2A). TEER values in monolayers treated with 0.9% DMSO vehicle were significantly decreased when media was changed at t = 0 min; no further changes in TEER were observed across the experimental duration. As previously documented, KCl application significantly decreased TEER [22] (Figure 2B,C). TEER values were also significantly reduced from baseline and compared to vehicle treatment at the end of the 15 min incubation period (0 min time-point) of either concentration of LEI-106 (Figure 2B) (LEI-106-650 µM vs. vehicle: *p* = 0.0017; LEI-106-1.3 mM vs. vehicle: *p* < 0.0001, as assessed by two-way ANOVA). At additional time-points, TEER was normalized; however, additional decrease in TEER values was detected at 2 and 3 h (Figure 2B) (2 h time-point: LEI-106-1.3 µM vs. vehicle: *p* = 0.0075; 3 h: LEI-106-650 µM vs. vehicle: *p* = 0.0254, LEI-106-1.3 mM vs vehicle: *p* < 0.022, as assessed by two-way ANOVA). The effect of LEI-106 on endothelial cell monolayer integrity over time was confirmed by AUC analysis (Figure 2C) (LEI-106-650 µM vs. vehicle: *p* = 0.009; LEI-106-1.3 mM vs. vehicle: *p* < 0.0001; LEI-106-650 µM vs. LEI-106-1.3 mM: *p* = 0.01, as assessed by one-way ANOVA with Tukey post-test).

In the siRNA transfection experiments, bEnd.3 cells were seeded on trans-well inserts 24 h post-transfection, then TEER was performed 72 h after the transfection. Silencing DAGLα by siRNA transfection significantly reduced TEER, compared to non-targeting control (Figure 2D) (DAGLα siRNA vs. non-targeting siRNA: *p* = 0.0003, as assessed by unpaired *t*-test, *n* = 5). Together, these TEER data provide evidence that the integrity of the endothelial monolayer is disrupted following pharmacological or genetic blockade of DAGLα.

In a separate experiment designed to establish whether the decreases in TEER induced by LEI-106 were a result of reduced 2-AG, cells received treatment with LEI-106 followed by media spiked with 2-AG (15 min, 600 pmol/well). Vehicle and LEI-106 were applied to the luminal side of the trans-well to simulate bloodstream delivery of drugs to the BBB; 2-AG was added to the abluminal side of the trans-well insert to mimic CNS production of 2-AG (Figure 2A). 2-AG treatment (600 pmol/well) in the abluminal chamber significantly increased TEER at individual time-points and as confirmed by AUC analysis (Figure 2E,F; 0 min time-point: LEI-106-650 µM vs. LEI-106-650 µM + 2-AG: *p* = 0.0002; LEI-106-1.3 mM vs. LEI-106-1.3 mM + 2-AG: *p* < 0.0001, as assessed by two-way ANOVA), (AUC: LEI-106-650 µM vs. LEI-106-650 µM + 2-AG: *p* = 0.0001; LEI-106-1.3 mM vs. LEI-106-1.3 mM + 2-AG: *p* < 0.0001, as assessed by one-way ANOVA). All measurements were repeated in triplicate (technical replicate) over three individual experiments (biological replicate). These data suggest that increasing 2-AG levels after DAGLα inhibitor restores endothelial monolayer electrical resistance. 

### 2.3. 2-AG Depletion Increases In Vitro ^14^C-Sucrose Transport

The next study examined ^14^C-sucrose transport across the bEnd.3 monolayer to determine if the observed TEER changes reflected functional changes in paracellular permeability (Figure 3A). Cells were treated with 2-AG at concentrations aligning with what has been recorded in vivo from the parenchyma (300 pmol/well–10 nmol/well; 600 µL/well) to determine basal responses to 2-AG on monolayer functionality (Figure 3B,C). Thirty min after treatment, 2-AG (300 pmol/well) increased ^14^C-sucrose transport across the monolayer (2-AG-300 pmol vs. vehicle: *p* = 0.0172, 2-AG at other doses vs. vehicle: *p* > 0.05, F(5,20) = 3.879, as assessed by one-way ANOVA with Bartlett’s test, *n* = 3–10/group); no other significant effects of 2-AG alone on bEnd.3 monolayer integrity were observed. Separate sets of cells were incubated in LEI-106 (1.3 mM), KCl (100 mM), or vehicle (0.9% DMSO) with or without 2-AG application (600 pmol/insert). LEI-106 (1.3 mM) significantly increased the ^14^C-sucrose detection in the abluminal chamber at 5- and 30-min time-points compared to vehicle control, indicating the presence of paracellular leak (Figure 3D,E) (5 min: LEI-106 vs. vehicle: *p* = 0.0033, 30 min: LEI-106 vs. vehicle: *p* = 0.0092, as assessed by unpaired *t*-test). We next investigated whether these paracellular breaches could be corrected with exogenously applied 2-AG. The application of 2-AG did not significantly influence the elevated sucrose uptake induced by LEI-106 at 5 min (Figure 3D) (LEI-106 vs. LEI-106 + 2-AG: *p* = 0.0797, as assessed by unpaired *t*-test). However, 2-AG treatment significantly mitigated the increase in sucrose uptake caused by DAGLα inhibition at 30 min (Figure 3E) (LEI-106 vs. LEI-106 + 2-AG: *p* = 0.05, as assessed by unpaired *t*-test). Data were obtained from three-four independent experiments using 4 trans-well inserts/group.

In the siRNA transfection experiments, bEnd.3 cells were seeded on trans-well-inserts 24 h post-transfection, then the ^14^C-sucrose assay was performed 72 h after the transfection. Transfection of DAGLα siRNA in bEnD.3 cells significantly increased the detection of ^14^C-sucrose in the abluminal chamber at 5 min, suggesting paracellular leak after DAGLα silencing. However, no significant change between DAGLα siRNA and control was observed at 30 min time-point (Figure 3F,G) (5 min: DAGLα siRNA vs. non-targeting siRNA: *p* = 0.0022, as assessed by unpaired *t*-test, *n* = 5; 30 min: DAGLα siRNA vs. non-targeting siRNA: *p* = 0.8204, as assessed by unpaired *t*-test, *n* = 5). Together, these data suggest that 2-AG signaling in endothelial cells functionally maintains barrier integrity and that loss of endogenous 2-AG tone promotes paracellular leak.

### 2.4. Immunocytochemistry of 2-AG Depleted Cells Reveals Morphological Changes

With confirmation that LEI-106 was reducing 2-AG within endothelial cells and that this loss functionally reduced monolayer paracellular integrity, we investigated possible underlying mechanisms, including cell viability, morphological changes, and junctional protein expression (Figure 4A). To test whether changes in barrier permeability reflect a loss of cell viability, an XTT cell viability assay was employed. Cells were treated with titrated doses of LEI-106 (100 nM, 10 µM, 100 µM, 325 µM, 650 µM, and 1.3 mM). No significant differences in oxidative capacity were noted at all LEI-106 doses (Figure 4B) (LEI-106 at any doses vs. vehicle: *p* > 0.05, as assessed by one-way ANOVA with Bartlett post-test, *n* = 4/group) suggesting DAGLα inhibition and loss of 2-AG does not play a role in bEnd.3 cell viability.

ICC revealed significant changes in endothelial cell morphology following LEI-106 (650 µM, 1.3 mM; Figure 4C,D). Total cell area was significantly decreased with a significant increase in nuclear: cytoplasmic (N:C) ratio following LEI-106. 2-AG treatment following LEI-106 significantly reversed these effects (Figure 4D) (LEI-106-650 µM vs. vehicle: *p* < 0.0001, LEI-106-1.3 mM vs. vehicle: *p* < 0.0001, as assessed by one-way ANOVA with Tukey post-test, *n* = 9–12/conditions) (LEI-106-650 µM vs. LEI-106-650 µM + 2-AG: *p* < 0.0001, LEI-106-1.3 mM vs. LEI-106-1.3 mM + 2-AG: *p* < 0.0001, as assessed by one-way ANOVA with Tukey post-test, *n* = 9–12/conditions) suggesting that endothelial-derived 2-AG plays a role in maintenance of endothelial cell structural integrity.

The immunoreactivity of VE-cadherin and claudin 5 was assessed to determine the impact of LEI-106 exposure on endothelial cell junctional proteins using the same cells as morphology experiments. LEI-106 (1.3 mM) significantly reduced VE-cadherin corrected total cell fluorescence (CTCF); no significant differences were detected at the lower dose (650 µM) compared to vehicle control (Figure 4E; LEI-106-650 µM vs. vehicle: *p* = 0.9052, LEI-106-1.3 mM vs. vehicle: *p* = 0.0233, as assessed by one-way ANOVA with Bartlett post-test, *n* = 9–12/condition). The administration of 2-AG increased the VE-cadherin CTCF, mitigating the loss of VE-cadherin expression after LEI-106 treatment (LEI-106-1.3 mM vs. LEI-106-1.3 mM + 2-AG: *p* = 0.005, as assessed by one-way ANOVA with Tukey post-test, *n* = 9–12/conditions). The blockade of DAGLα with LEI-106 at either 650 µM or 1.3mM did not induce significant changes in the CTCF of claudin-5 compared to the vehicle control (Figure 4F) (LEI-106-650 µM vs. vehicle: *p* = 0.8982, LEI-106-1.3 mM vs. vehicle: *p* = 0.6161, as assessed by one-way ANOVA with Tukey post-test, *n* = 9–12/condition). These data indicate that the concentrations of LEI-106 utilized in this study do not reduce endothelial cell viability and that the alterations observed following LEI-106 treatment reflect changes to endothelial morphology and expression of the adherens junctional protein, VE-cadherin.

### 2.5. Endothelial Tight Junction Protein Detection Is Modulated Following DAGLα Inhibition 

The next series of experiments sought to validate the ICC observations of VE-cadherin and claudin 5 immunofluorescence by performing Western blot analysis following LEI-106 treatment or gene editing (Figure 5A). Cell lysates harvested after LEI-106 (650 µM, 1.3 mM; 15 min) were first probed for VE-cadherin. VE-cadherin was detected at 130 kDa in all treatment conditions. In LEI-106-treated cell lysates, a notable fragment at approximately 100 kDa was also detected, indicating that VE-cadherin had undergone a fragmenting event following DAGLα inhibition. The treatment of LEI-106 at 650 µM dose did not cause a significant change in the detection of VE-cadherin main 130kDa band compared to vehicle control (Figure 5B) (vehicle vs. LEI-106, 650 µM: *p* = 0.3565 as assessed by one-way ANOVA with Bartlett’s test, *n* = 8–12/group), but it significantly increased the detection of 100kDa fragment (vehicle vs. LEI-106, 650 µM: *p* = 0.0014 as assessed by one-way ANOVA with Bartlett’s test, *n* = 8–12/condition). The higher dose of LEI-106 (1.3 mM) significantly reduced the detection of VE-cadherin main band and increased the detection of the fragmented one compared to vehicle control (Figure 5B) (main band: vehicle vs. LEI-106, 1.3 mM: *p* < 0.0001 as assessed by one-way ANOVA with Bartlett’s test; fragment: vehicle vs. LEI-106, 1.3 mM: *p* < 0.0001 as assessed by one-way ANOVA with Bartlett’s test, *n* = 8–12/group). Interestingly, the application of 2-AG further decreased the detection of main VE-cadherin band, but it did not significantly influence the fragmentation compared to corresponding controls (Figure 5B) (main band: LEI-106-650 µM vs. LEI-106-650 µM + 2-AG: *p* = 0.0092, LEI-106, 1.3 mM vs. LEI-106, 1.3 mM + 2-AG: *p* = 0.011 as assessed by one-way ANOVA with Bartlett’s test; fragment: LEI-106-650 µM vs. LEI-106-650 µM + 2-AG: *p* = 0.9665, LEI-106-1.3 mM vs. LEI-106-1.3 mM + 2-AG: *p* = 0.0769 as assessed by one-way ANOVA with Bartlett’s test, *n* = 8–12/condition). The expression of VE-cadherin was also assessed at 72 h after siRNA transfection in bEnD.3 cells. The transfection of DAGLα-specific siRNA significantly decreased the detection of VE-cadherin compared to the non-targeting control (Figure 5C) (DAGLα siRNA vs. non-targeting control siRNA: *p* < 0.0001, as assessed by unpaired *t*-test, *n* = 8/group); this strategy did not lead to detection of the VE-cadherin fragment. These data, taken with the ICC data, suggest regulation of endothelial-derived 2-AG regulates the detection of VE-cadherin in bEnd.3 cells.

Claudin 5 was detected in all cell lysates at 18kDa (Figure 5D). WB analysis of claudin-5 signal was significantly decreased after LEI-106 treatment at the higher dose (vehicle vs. LEI-106-650 µM: *p* = 0.0759; vehicle vs. LEI-106-1.3 mM: *p* = 0.0076 as assessed by one-way ANOVA with Tukey’s test, *n* = 8–10/group). The application of 2-AG did not cause significant changes compared to the corresponding controls (LEI-106-650 µM vs. LEI-106-650 µM + 2-AG: *p* = 0.9870, LEI-106-1.3 mM vs. LEI-106-1.3 mM + 2-AG: *p* = 0.6535 as assessed by one-way ANOVA with Tukey’s test, *n* = 8–10/condition). No significant difference in the detection of claudin-5 was observed after the transfection of DAGLα-specific siRNA either (Figure 5E) (DAGLα siRNA vs. non-targeting control siRNA: *p* = 0.9666, as assessed by unpaired *t*-test, *n* = 6/group). Together with the ICC results, these data indicate that the pharmacological inhibition of DAGLα has a limited impact on the detection of claudin 5 in bEnd.3 cells.

### 2.6. DAGLα Inhibition Increases Cellular Calcium Detection with Engagement of Protein Kinase C (PKC) and Rho-Kinase Pathways

VE-cadherin is a calcium-dependent junctional protein, and prior reports indicate that VE-cadherin is fragmented by a calcium-dependent kinase, calpain [23]. The next experiments sought to determine whether the fragment of VE-cadherin detected after LEI-106 treatment resulted from increasing intracellular calcium within bEnd.3 cells and calpain engagement. The first study used calcium imaging of bEnd.3 cells with the dye Fura-2, AM. Test cells underwent a 2-min baseline observation in buffer, followed by 2-min treatment with LEI-106 (650 µM, 1.3 mM), vehicle (0.9% DMSO), and a subsequent 2-min washout phase. In a separate experiment, cells received buffer spiked with 2-AG (600 pmol/coverslip) during the washout phase (Figure 6A). bEnd.3 cells treated with LEI-106 at both doses displayed significant increases in intracellular calcium compared to vehicle control (Figure 6B) (LEI-106-650 µM vs. vehicle: *p* < 0.0001, LEI-106-1.3 mM vs. vehicle: *p* < 0.0001, as assessed by one-way ANOVA with Tukey post-test). Treatment with 2-AG spiked buffer following LEI-106 significantly decreased detection of intracellular calcium compared to buffer alone (Figure 6C) (LEI-106-650 µM vs. LEI-106-650 µM + 2-AG: *p* < 0.0001, LEI-106-1.3 mM vs. LEI-106-1.3 mM + 2-AG: *p* < 0.0001, as assessed by one-way ANOVA with Tukey post-test). To determine if this partial 2-AG effect was mediated by cannabinoid receptors (CB_1_R and CB_2_R), we repeated the experiment with the non-selective synthetic agonist WIN-55,212. Application of WIN-55,212 after LEI-106 reduced the calcium signal compared to LEI-106 alone (LEI-106-1.3 mM vs. LEI-106-1.3 mM+ WIN-55,212: *p* < 0.0001, t(14.30) = 856, as assessed by unpaired *t*-test). Together, these data indicate that LEI-106 treatment increases intracellular calcium concentration in a dose-dependent manner over time, which can be partially corrected with exogenous 2-AG supplementation and the non-selective CB_1_R/CB_2_R agonist WIN-55,212.

Having established that intracellular calcium levels are increased after treatment with LEI-106, we tested if the calcium-dependent kinase, calpain, played a role in the cleavage of VE-cadherin caused by DAGLα blockade. bEnD.3 cells were treated with LEI-106 in combination with calpain inhibitor calpeptin (10 µM, 30 min pretreatment). The calpain inhibitor did not significantly mitigate the fragmentation of VE-cadherin caused by LEI-106 treatment (Figure 6D) (LEI-106-650 µM vs. LEI-106-650 µM + calpeptin: *p* > 0.9999, as assessed by one-way ANOVA with Tukey post-test, *n* = 5–6/condition) suggesting that calpain did not play a role in the cleavage of VE-cadherin caused by DAGLα inhibition. 

Previous work has shown as well that Ca-signaling, along with activation of protein kinase C (PKC), has a critical role in mediating the disruption of VE-cadherin [24]. Therefore, the activity of PKC was determined in LEI-106-treated bEnD.3 cells. PKC activity, as assessed by ELISA, showed that LEI-106 treatment at both doses (15 min) significantly increased the PKC activity compared to vehicle control (Figure 6E) (LEI-106-650 µM vs. vehicle: *p* = 0.0289, t(8) = 2.658; LEI-106-1.3 mM vs. vehicle: *p* = 0.0099, as assessed by unpaired *t*-test, *n* = 5/group). In order to test how PKC inhibitor can affect the fragmentation of VE-cadherin caused by LEI-106, bEnD.3 cells were pretreated with PKC inhibitor, G06938 (100 µM), 30 min before DAGLα inhibition. The PKC inhibitor did not cause significant changes in the fragmentation of VE-cadherin (Supply Appendix A) (LEI-106-650 µM vs. LEI-106-650 µM + G06938: *p* = 0.9999, LEI-106-1.3 mM vs. LEI-106-1.3 mM+ G06938: *p* = 0.9548, as assessed by one-way ANOVA with Tukey post-test, F(5,18) = 21.88, *n* = 4/condition).

To further investigate the possible signaling mechanism underlying how DAGLα inhibition resulted in VE-cadherin fragmentation, the pan-Rho-kinase inhibitor SR3677 was tested in combination with LEI-106. bEnD.3 cells were treated with SR3677 (100 nM) 30 min prior to LEI-106 (650 µM), then subjected to Western immunoblotting to probe for VE-cadherin. Rho-kinase inhibitor significantly mitigated the fragmentation of VE-cadherin caused by DAGLα inhibitor (Figure 6F) (LEI-106-650 µM vs. LEI-106-650 µM + SR3677: *p* < 0.0001, as assessed by one-way ANOVA, *n* = 8–10/condition). Together, these data suggest a role for increased intracellular Ca-signal, elevated PKC activity, and the engagement of the Rho-kinase pathway in the effect of DAGLα inhibition on VE-cadherin.

To further elucidate the effect of DAGLα inhibition in endothelial cells, an unbiased phospho-proteomics screen was performed on bEnd.3 cells treated with either vehicle or the DAGLα inhibitor LEI-106 (Figure 7A). Mass spectrometry analysis of the phospho-peptides purified from the bEnd.3 lysates detected a total of 24,505 phospho-peptide ions (Appendix A), of which 8166 possessed significantly different abundances between the two groups (Figure 7B—VOLCANO). Unbiased principal component analysis (PCA) of the significantly affected phospho-peptide ions clustered the samples according to treatment, supporting the strength of the effect of LEI-106 inhibitor on changes in phosphorylation (Figure 7C—PCA). While within-group variance along component 2 accounted for only 7.7% of the variance, the LEI-106-treated samples did exhibit more pronounced phospho-peptide ion-abundance heterogeneity compared to the vehicle controls. Hierarchical protein clustering, heatmap visualization, and protein cluster profiles of the 8166 significantly affected phospho-peptide ions confirmed that samples within groups cluster together, supporting the reproducibility of the findings (Figure 7D—HEAT MAP). Of the 8166 significantly affected phospho-peptide ions, 5589 (68.4%) exhibited a greater abundance after the LEI-106 treatment (Appendix A); these occurred across 1335 distinct proteins. The top five proteins with phospho-peptides, as identified by ANOVA, were ARC1B, ATG9Am BAZ1B, DDX21, and DZIP3 (Appendix A). Gene Ontology enrichment analysis of the proteins displaying significant changes in phospho-peptide ion abundance after LEI-106 treatment was performed for biological processes, cell compartment, and molecular function; Reactome pathways were also assessed (Figure 7E–H, respectively). Biological processes with increased phospho-peptide detection revealed 459 functional clusters, including significantly increased phosphorylation of proteins involved in mRNA processing and binding. Molecular function clustering revealed 12 distinct clusters with small GTPase activity and RNA binding within the top hits. Reactome analysis identified 63 unique pathways enriched after LEI106 treatment, including phosphorylation of proteins involved in mRNA processing and binding, as well as pathways regulating mRNA splicing, RUNX1-mediated gene regulation, and estrogen-dependent gene expression. In contrast, 2577 phospho-peptide ions were decreased by LEI106 compared to vehicle treatment (Figure 7D—**HEAT MAP**) across 766 individual proteins. The top five proteins with phospho modifications included HS0B, STIM2, PRP4B, PRC2C, and CBX1 (Appendix A). Related to the pharmacological experiment in Figure 6, ROCK 2 had sites (^1121-^SQLQALHIGMDSSSIGSGPGDAEPDDGFPESR^−1152^ that were significantly decreased by LEI106 (*p* < 0.001, Appendix A); these fall within the pleckstrin homology and cysteine-rich domains. The number of functional clusters identified for biological processes, molecular function, and Reactome pathways were 339, 7, and 43, respectively. Proteins displaying a significant loss in phospho-peptide ion abundance after LEI-106 treatment revealed gene ontology enrichment for cytoskeletal organization, activation of GTPase activity, including Rho isoform cycling, actin binding, focal adhesions, and VEGFA-VEGFR2 pathways. Together, these data suggest that the blockade of DAGLα significantly alters the functional states of brain endothelial cells.

Proteomics data also suggest changes in another tight junction protein, Zonula occludens-1 (ZO-1), after DAGLα inhibition. ZO-1, in contrast to transmembrane tight junction proteins, like claudin-5 and VE-cadherin, is located intracellularly, forming a scaffold between the transmembrane proteins and the actin cytoskeleton. In order to confirm proteomics findings, a Western experiment was performed to detect ZO-1 expression after LEI-106 treatment. The blockade of DAGLα significantly reduced the detection of ZO-1 compared to vehicle control (Figure 7I) (vehicle vs. LEI-106-650 uM: *p* = 0.034, vehicle vs. LEI-106-1.3 mM: *p* = 0.0004, as assessed by one-way ANOVA with Tukey post-test, *n* = 3-4/condition).

Together, these data suggest that disruption of 2-AG homeostasis by DAGLα inhibition significantly alters the phospho-proteome of bEnd.3 cells.

## 3. Discussion

While it is accepted that (endo)cannabinoids can enhance the integrity of the blood–brain barrier (BBB) in pathologic states, the exact mechanisms by which this occurs are not yet fully understood [21]. The above studies aimed to gain insight into the role of DAGLα, the 2-AG synthetic enzyme, and endothelial-derived 2-AG in BBB function using a bEnd.3cell monolayer model. Using both pharmacological inhibition and siRNA manipulation to decrease DAGLα function, endothelial-derived 2-AG was decreased. This loss of 2-AG compromised bEnd.3 monolayer paracellular integrity by increasing the intracellular calcium-signaling activity of protein kinase C and fragmenting VE-cadherin. The reduced functional expression of VE-cadherin was dependent on Rho-kinase but not calpain. Thus, constitutive production of 2-AG in brain endothelium maintains BBB homeostasis by regulating junctional integrity through phosphorylation and not protein internalization. Figure 8 summarizes all our findings.

Evidence has increased in the last decade of decreased endocannabinoid tone in neuropathologies, including dementia, traumatic brain injury, and migraine [15,25,26,27]. These neuropathologies also disrupt BBB integrity, and methods to increase endocannabinoid tone have shown promise as therapeutic interventions, suggesting a role for the endocannabinoid system in BBB homeostasis [2,15,18,19,20,26,27,28]. The current data using selective DAGLα inhibition with LEI-106 and siRNA targeting of DAGLα showed that significant decreases in endothelial-derived 2-AG levels, but not AEA or DAG, in endothelial cell lysates impaired endothelial cell monolayer integrity as indicated by reduced TEER and increased uptake of ^14^C-sucrose across the monolayer. These data are consistent with prior in vivo evidence of 2-AG mitigating BBB impairment, further supporting the role of 2-AG tone in endothelial cells in BBB homeostasis [18,19,20].

Breaches of BBB integrity during CNS disorders and diseases have been attributed to endothelial cell death [29,30,31], changes in morphology (e.g., stress fiber formation) and junctional protein instability [32,33]. Following LEI-106, cell viability was not changed, suggesting that acute depletion of 2-AG did not facilitate cell death. However, ICC showed cell morphology shifted away from a confluent monolayer and toward isolated cell clusters 15 min after LEI-106. Co-treatment of LEI-106 with exogenous 2-AG prevented these morphological shifts, suggesting that endothelial cells require tonic 2-AG tone to maintain shape and functional integrity.

In addition to cytoskeletal remodeling, the instability of junctional proteins is attributed to multiple mechanisms, including protein cleavage and internalization, as well as post-translational modification [34,35,36]. Of note, both cytoskeletal remodeling and protein cleavage have calcium-dependent and -independent mechanisms. Our data supports a role for endothelial-derived 2-AG in regulating intracellular Ca^2+^ levels that could be reversed by 2-AG application. In both ICC and Western blot analysis of bEnd.3 cell lysates, detection of the TJ protein, VE-cadherin was decreased with increased fragmentation after pharmacological blockade of DAGLα. Reduced detection of claudin 5 was observed after LEI-106 treatment in Western experiments. Detection of VE-cadherin was significantly decreased in cells transfected with DAGLα-targeting siRNA; however, neither the fragmentation of VE-cadherin nor the reduced expression of claudin-5 was observed after genetic inhibition of DAGLα, suggestive of differing regulation of junctional proteins with loss of protein vs pharmacological inhibition. The time-line difference between pharmacological (15 min) and siRNA manipulation (72 h) can also provide a possible rationale since the pharmacological blockade can induce rapid changes, whereas the mRNA silencing does not immediately impact protein synthesis; the detection difference in VE-cadherin main band and fragment after LEI-106 but not DAGLα silencing exemplifies this idea. With respect to claudin-5, the effect of LEI-106 might reflect an off-target effect since no changes were observed when the siRNA was introduced. The observed reduction in the detection of ZO-1 after LEI-106 can also contribute to the permeability change showed in functional assays, but further experiments need to be conducted to answer how the changes in ZO-1 and VE-cadherin after DAGLα inhibition are related to each other. Interestingly, exogenous 2-AG application was not able to diminish the effect of DAGLα inhibition on junctional protein detection, suggesting these changes in protein function were calcium-independent. Thus, endothelial-derived 2-AG plays a role in BBB tone in both Ca^2+^-dependent and -independent manners.

VE-Cadherin plays a critical role in maintaining cell-cell adhesions at the BBB, and complete loss of functional VE-cadherin is lethal [37]. In pathological states, multiple mechanisms are linked to the regulation of VE-cadherin functional expression, including fragmentation and internalization via calpain. Our results indicate increased Ca^2+^ influx in bEnD.3 cells after pharmacological inhibition of DAGLα, but not the involvement of calpain in the effect of LEI-106 on VE-cadherin. A recent review paper discusses the role of calcium in barrier integrity, highlighting that calcium influx triggers adherens junction disassembly and facilitates endothelial permeability [38]. However, this review paper focuses on the changes in barrier integrity during inflammation; another review article written by Brown and Davis also describes that calcium can regulate functional expression of the tight junction proteins on cerebral microvessels during stroke [39]. Based on our data, the loss of 2-AG via DAGLα inhibition did facilitate calcium influx in bEnD.3 cells, indicating a homeostatic role of 2-AG in these processes.

A study published by Sandoval et al. showed that calcium signaling, along with PKC activity, has a critical role in mediating the disruption of VE-cadherin in human umbilical vein endothelial cells [24]. Therefore, to further explore the possible mechanism underlying the effect of DAGLα, we tested the involvement of protein kinase C (PKC)-related pathways. The activity of PKC in bEnD.3 cells was significantly increased after LEI-106 treatment; however, the pretreatment with PKC inhibitor G06938 did not prevent the fragmentation of VE-cadherin after DAGLα blockade. In contrast, the application of ROCK inhibitor SR3677 mitigated the fragmentation of VE-cadherin caused by LEI-106. This data aligns well with former work published by Li et al. [40]. Their results showed that ROCK pathway inhibitor Y27632 increased VE-cadherin expression in human umbilical vein endothelial cells. The proteomics data also suggests the alteration of Rho-pathways, including Rho GTPase-activating proteins (Rho-GAPs) and Rho guanine nucleotide exchange factors (Rho-GEFs). GAPs and GEFs are highly expressed in endothelium, and they play a role in endothelial cell adhesion and vascular homeostasis [41]. A complete understanding of the connection between 2-AG signaling and Rho-pathways in endothelial cells requires future experiments. The involvement of other pathways, including mRNA splicing, RUNX1-mediated gene regulation, and estrogen-dependent gene expression in the effect of DAGLα blockade on endothelial cells is indicated by the phospho-proteomics screening. Follow-up experiments to re-validate those findings and further exploration of 2-AG signaling in endothelial cells are called for subsequent work.

It is known that the endocannabinoid system is involved in several aspects of intestinal physiology, including epithelial barrier function [42]. The structural resemblance between blood–brain barrier and gut barrier raises the question of whether the endocannabinoid system has similar protective role in the gastrointestinal tract. A recent paper published by Wiley and DiPatrizio found that the 2-AG level in the intestinal epithelium was significantly reduced in rodent models of diet-induced obesity [43]. This was associated with disrupted gut barrier functions, reduced DAGL activity, and changes in the expression of genes controlling tight junction, including ZO-1 and occludin. Those findings support the beneficial role of the endocannabinoid system in endothelial function and maintenance of tight junction, independently from the type of tissue. Moreover, there is a possibility that similar subcellular signaling in brain and gut endothelial is driven by the ECB system. Future studies need to be conducted to fully elucidate this possibility.

Limitations: Multiple limitations are present within this study. While it was demonstrated that the in vitro dosing of LEI-106 did not decrease cell viability, the doses used in this study, chosen from Levine et al., 2021 [25], were intended to represent maximal bioavailability and likely do not reflect the true delivery of LEI-106 to the endothelial cells of the NVU. Future work is necessary to examine lower doses of LEI-106 in cell cultures. The mechanism by which LEI-106 increases intracellular calcium remains elusive as well. Some of the effects of DAGLα blockade on the endothelial integrity were mitigated by exogenous application of 2-AG. It is known that 2-AG can act at cannabinoid-1 and cannabinoid-2 receptors as well; therefore, the role of each receptor in the maintenance of endothelial integrity needs to be examined in future studies using selective ligands. The reliance on Western blot and ICC on antibody binding limits the results of these studies as well. Finally, the in vitro culture of endothelial cells is not able to completely mimic the complexity of BBB in vivo; therefore, it is necessary to evaluate the molecular observations presented in this study, including levels of calcium, VE-cadherin, and claudin 5 following in vivo administration of LEI-106.

## 4. Materials and Methods

### 4.1. Drugs

LEI-106, WIN-55,212, calpeptin, G06983, SR3677, 2-AG, 2-AG-d5, and AEA-d4 were purchased from Cayman Chemicals (Ann Arbor, MI, USA). Cells received LEI-106 (650 µM, 1.3 mM) dissolved in 0.9% DMSO in cell media 15 min prior to analysis. The doses of LEI-106 used for cell assays were derived as equimolar doses to previous in vivo assays, assuming 50% and 100% bioavailability. Titrated doses of LEI-106 in 0.9% DMSO (100 nM, 10 µM, 100 µM, 325 µM, 650 µM, 1.3 mM, 10 mM) were utilized to assess for cell viability. 2-AG was dissolved in 0.9% DMSO in cell media and was administered to cells in 600 pmol dose. Calpeptin (10 µM), G06983 (100 nM), and SR3677 (100 nM) were applied 30 min before LEI-106 treatment.

### 4.2. Cell Culture

Murine brain endothelial cells, bEnd.3 (CRL-2299, ATCC) were cultured in DMEM (Gibco, Sigma Aldrich, St Louis, MO, USA 11995-065), supplemented with 2 mM L-glutamine (ThermoScientific, 25030081), 10% fetal bovine serum (Gibco, 10082139), and penicillin (100 UI/mL)-streptomycin (100 μg/mL) (Invitrogen, 15140122). Cells were cultured at 37 °C in a humidified 5% CO_2_/95% air atmosphere and were grown to 80% confluence in all experiments. Twenty-four hours before treatment, the media of bEnd.3 cells were changed to astrocyte-conditioned media (ACM) harvested from confluent C8-D1A flasks. The C8-D1A cells (CRL-2541, ATCC) were cultured in DMEM (Gibco, 11995-065), supplemented with 10% fetal bovine serum and penicillin-streptomycin. To mimic CSD event in vitro, bEnd.3 cells were treated with 100 mM KCl for 5 min, which is a typical condition to evoke potassium-triggered spreading depolarization in live brain slices [44].

### 4.3. Liquid Chromatography-Mass Spectrometry (LC-MS)

Cellular samples for LC-MS were purified by organic solvent extraction with a protocol modified from Wilkerson et al. [45]. Then, 2 × 10^6^ cells were seeded into 10cm^2^ culture dishes and allowed to grow for 7 days. On the day of processing, cells were treated and then collected via cell scraping and placed into microcentrifuge tubes. Samples were then centrifuged (1000× *g*, for 10 min at 4 °C), and the supernatant was removed. The resulting pellet was then weighed and homogenized in 1 mL of chloroform/methanol (2:1 *v*/*v*) supplemented with phenylmethylsulfonyl fluoride (PMSF) at 1 mM final concentration to inhibit the degradation by endogenous enzymes. Homogenates were then mixed with 0.3 mL of 0.7% *w*/*v* NaCl, vortexed, and then centrifuged for 10 min at 3200× *g* at 4 °C. The aqueous phase plus debris were collected and extracted two more times with 0.8 mL of chloroform. The organic phases from the three extractions were pooled, and an internal standard was added to each sample. Mixed internal standard solutions were prepared by serial dilution of AEA-d4 and 2-AG-d5 in 80% acetonitrile. The organic solvents were evaporated under nitrogen gas. Then, 6 µL of 30% glycerol in methanol per sample was added before evaporation. Dried samples were reconstituted with 0.2 mL of chloroform and mixed with 1 mL of ice-cold acetone to precipitate proteins. The mixtures were then centrifuged for 5 min at 1800× *g* at 4 °C. The organic layer of each sample was collected and evaporated under nitrogen.

Analysis of 2-AG and AEA was performed on an Ultivo triple quadrupole mass spectrometer combined with a 1290 Infinity II UPLC system (Agilent, Palo Alto, CA, USA). The instrument was operated in electrospray positive mode with a gas temperature of 150 °C at a flow of 5 L/min, nebulizer at 15 psi, capillary voltage of 4500 V, sheath gas at 400 °C with a flow of 12 L/min and nozzle voltage of 300 V. Transitions monitored were 348.3 → 287.3 and 62, 352.3 → 287.4 and 65.9, 379.3 → 287.2 and 269.2, and 384.3 → 287.2 and 296.1 for AEA, AEA-d4, 2-AG, and 2-AG-d5. The first fragment listed was used for quantification, and the second fragment was used for confirmation. The first 3 min of analysis time was diverted to waste. Chromatographic separation was achieved using an isocratic system of 21% 1 mM ammonium fluoride and 79% methanol on an Acquity UPLC BEH C-18 1.7u 2.1 × 100 mm column (Waters, Milford, MA, USA) maintained at 60 °C. After each injection, the column was washed with 90% methanol for 1 min, then re-equilibrated for 5 min prior to the next injection. Samples were maintained at 4 °C. Mixed calibration solutions were prepared by serial dilution of AEA and 2-AG stock solutions in 80% acetonitrile. Calibration curves were prepared for each analysis by adding 10 µL internal standard solution to 20 µL standard solution. Prior to sample analysis, 200 µL of 80:20 acetonitrile:water was added to dried samples, which were then vortexed and sonicated. The samples were centrifuged at 15,800× *g* at 4 °C for 5 min. The supernatant was transferred to autosampler vials, and 5 µL was injected for analysis.

### 4.4. siRNA Transfection of bEnD.3 Cells to Silence DAGLα Gene

bEnd.3 cells were seeded onto 6-well plates at 200,000 cells/well density and cultured in regular media for 24 h. On the day of the experiment, the cells were incubated in FBS and antibiotic-free DMEM for 2 h before the transfection. Double-stranded siRNA targeting DAGLα was purchased from ThermoScientific (Waltham, MA, USA; Silencer^®^ Select Pre-designed siRNA, #s114215). The sequences targeting DAGLα are the following: CCAAGUACCUCGACCUCAAtt (sense) and UUGAGGUCGAGGUACUUGGtg (antisense). Non-targeting control siRNA (ThermoScientific, Waltham, MA, USA #4390843) was applied as a negative control. Lipofectamine RNAiMAX (ThermoScientific, #13778) was used as a transfection reagent. siRNAs were diluted in Opti-MEM for 50 pmol/well as the final concentration. The cells were incubated with the siRNA-Lipofectamine mixture in Opti-MEM for 4 h, and then the media was changed to the regular one. Subsequent experiments, including Western blotting, TEER, sucrose assays, and LC-MS, were performed 72 h post-transfection. For TEER and sucrose assays, the transfected cells were cultured on trans-well inserts from 24 h post-transfection. The transfection efficacy was validated by Western blotting probing for DAGLα (antibody: Cell Signaling, Danvers, MA, USA #13626S) and α-tubulin as loading control. In a separate experiment, the linear relationship between the different amounts of loaded protein and the signal of the DAGLα antibody was measured using serial dilution of naïve bEnD.3 cell lysate.

### 4.5. Trans-Endothelial Electrical Resistance (TEER)

TEER experiments were conducted as described in Liktor-Busa et al. to capture possible breaches in the active and passive transport [46]. Briefly, bEnd.3 cells were seeded on collagen-coated trans-well inserts (Corning, Sigma Aldrich, St. Louis, MO, USA) at 6.0 × 10^4^ cells/cm^2^ density. The regular bEnd.3 media was used on the luminal side, and astro-conditioned media (ACM) was added to the abluminal side to facilitate barrier functions. ACM was collected as described above. LEI-106 (650 µM or 1.3 mM, 0.9% DMSO in Media) with or without 2-AG (600 pmol) was applied to the luminal side of the trans-well insert. KCl pulse (100 mM KCl in media) was added to the abluminal side. Baseline measurements were taken before any treatment, then right after treatment (0 time-point), and then 10, 20, 30 min, 1, 2, 3, and 24 h post-treatment via 2-electrode, chopstick method (EVOM2). All measurements were repeated in triplicate over three individual experiments in a temperature-controlled environment (at 37 °C).

### 4.6. In Vitro ^14^C-Sucrose Transport 

bEnd.3 cells (CRL-2299, ATCC, Manassas, VA USA) were cultured on collagen-coated trans-well membranes as described in the former section and in our former paper [46]. LEI-106 (1.3 mM, 0.9% DMSO in Media) was added to the luminal side of the membrane 15 min prior to analysis. Following pretreatment, luminal media was replaced with fresh bEnd.3 media containing ^14^C-Sucrose (0.25 uCi/mL). Abluminal media was collected at 5 min and 30 min post-treatment and analyzed for disintegrations per minute (dpm) using a model 1450 liquid scintillation counter (PerkinElmer, Shelton, CT, USA). The 5 min sample collection represented the sucrose transport in the first 5 min of the experiment. The 30 min sample collection represented the sucrose transport between 6–30 min. In a separate experiment, following pretreatment, 2-AG (600 pmol) was supplemented into the abluminal media to simulate increased eCB signaling. Experiments were carried out in triplicate, with 4 trans-well inserts/group.

### 4.7. Immunocytochemistry (ICC)

bEnd.3 cells were grown on collagenated glass coverslips to confluence and treated as described above in “*Materials and Methods: Drugs*”. Cells were then washed with phosphate buffer saline (PBS) twice before fixing them with 1% Paraformaldehyde (PFA) for 15-min followed by a 1 h incubation in 10% BSA as a blocking agent. Following PBS washes, the cells were incubated in primary antibodies (Rabbit anti-VE-cadherin, Cell Signaling 2500S; Mouse anti-Claudin 5, Invitrogen, Waltham, MA, USA 35-2500) overnight in a 4 °C room. After washing with PBS, the cells were incubated in secondary antibodies (Goat anti-Rabbit AlexaFluor 488, Invitrogen A-11008; Donkey anti-Mouse AlexaFluor 647, Invitrogen, Waltham, MA, USA, cat# A-31571) at a 1:500 dilution in blocking buffer for 1 h at room temperature. The coverslips were mounted after additional PBS washes using Prolong Gold Antifade with DAPI (ThermoScientific, P36941) mounting media and were then dried and sealed. The images of the slides were captured with an ECHO Revolve Microscope (ECHO, San Diego, CA, USA) at 40× magnification. Four slides were generated per condition, with three images obtained for each slide. The corrected total cell fluorescence (CTCF) was calculated as the Integrated Density—(Area of selected cell × Mean fluorescence of background readings) for five cell ROIs/image and five background readings/image. All values determined in FIJI ImageJ as previously described [47]. 

### 4.8. Western Immunoblotting

bEnd.3 cells were cultured in 6-well plates in regular media. Twenty-four hours before the experiment, the media was changed to ACM, as described above. On the day of the experiment, cells were treated with LEI-106 (650 µM or 1.3 mM) for 15 min, then washed with ice-cold PBS. In a separate experiment, cells underwent DAGLα blockade via siRNA transfection, as described above. The cells were lysed in ice-cold lysis buffer (20 mM Tris-HCl (pH 7.4), 50 mM NaCl, 2 mM MgCl_2_ hexahydrate, 1% (*v*/*v*) NP40, 0.5% (*w*/*v*) sodium deoxycholate, 0.1% (*w*/*v*) SDS supplemented with protease and phosphatase inhibitor cocktail (Halt Protease and Phosphatase inhibitor cocktail, ThermoScientific, #78441). The lysate was centrifuged at 12,000× *g* for 10 min at 4 °C, and then the supernatant was collected. The protein content of the supernatant was determined by BCA assay. Then, 25 µg of total protein was loaded into TGX precast gels (4–20% Criterion^TM^, BioRad, Boston, MA, USA) and transferred to nitrocellulose membrane (Amersham^TM^ Protran^TM^, GE Healthcare, Providence, RI, USA). After transfer, the membrane was blocked at room temperature for 30 min in a blocking buffer (5% dry milk in Tris-buffered saline (TBS)). The following primary antibodies were used: NPAS4 (ThermoScientific, MA5-27592, 1:500), VE-cadherin (ThermoScientific, 36-1900, 1:500), claudin-5 (ThermoScientific, 35-2500, 1:500), and α-tubulin (Cell Signaling, Danvers, MA, USA, Cat# 3873S, 1:10,000). The primary antibodies were diluted in 5% BSA in TBST (Tris-buffered saline with Tween 20). The membrane was incubated in diluted primary antibodies for 48 h at 4 °C. The membrane was then washed three times in TBST for 5 min each followed by incubation with IRDye^®^ 800CW Donkey anti-Rabbit IgG Secondary Antibody (Li-Cor, Lincoln, NE, USA, cat # 926-32213) and IRDye^®^ 680RD Donkey anti-Mouse IgG Secondary Antibody (Li-Cor, 926-68072) in 5% milk in TBST for 1 h rocking at room temperature. The membrane was washed again two times for five minutes each in TBST. TBS buffer was used for the last washing step. The membrane was imaged with an Azure Sapphire laser imager (Azure Biosystems, Dublin, CA, USA). Un-Scan-It 6.1 software (Silk Scientific Inc., Provo Utah, USA) was used for quantification.

### 4.9. Calcium Imaging

Calcium imaging was performed as previously described [48]. bEnd.3 cells grown on coverslips were incubated at 37 °C with 3 µM Fura-2AM (Cat#F-1221; Life Technologies, Waltham, MA USA) stock solution prepared at 1 mM in DMSO, 0.02% pluronic acid, Cat#P-3000MP; Life Technologies) for 30 min (K_d_ = 25 µM, λ_ex_ 340, 380 nm/λ_emi_ 512 nm) to follow changes in intracellular calcium in DMEM (Gibco, 11995-065), supplemented with 2 mM L-glutamine (ThermoScientific, 25030081), 10% fetal bovine serum (Gibco, 10082139), and penicillin (100 UI/mL)-streptomycin (100 μg/mL) (Invitrogen, Waltham, MA, USA, Cat # 15140122). All calcium imaging experiments were done at room temperature (~23 °C). Treatments were performed following 2 min baseline with 1X Phosphate Buffer Solution containing 2 mM Calcium Chloride. Cells were treated with vehicle (0.9% DMSO in 1x PBS), KCl (100 mM), or LEI-106 (650 µM, 1.3 mM, 2 min or 15 min) followed by 2 min washout with a phosphate buffer with and without 2-AG (600 pmol/coverslip). In a separate experiment, cells underwent DAGLα blockade via siRNA transfection, described below. Fluorescence imaging was performed with an inverted microscope, Nikon Eclipse T*i*-U (Nikon Instruments Inc., Melville, NY, USA), using objective Nikon Super Fluor MTB FLUOR ×10 0.50 and a photometrics cooled CCD camera CoolSNAP ES^2^ (Roper Scientific, Planegg, Germany) controlled by NIS Elements software (version 4.20; Nikon Instruments). The excitation light was delivered by a Lambda-LS system (Sutter Instruments, Novato, CA, USA). The excitation filters (340 ± 5 and 380 ± 7 nm) were controlled by a Lambda 10 to 2 optical filter change (Sutter Instruments). Fluorescence was recorded through a 505-nm dichroic mirror at 535 ± 25 nm. The images were taken every 10 s during the time-course of the experiment, using the minimal exposure time that provided acceptable image quality to minimize photobleaching and phototoxicity. The changes in calcium were monitored by following the ratio of F_340_/F_380_, calculated after subtracting the background from both channels. The region of interest (ROI) tool was used to define 150 cells per coverslip to measure calcium signaling in each individual cell, where one ROI covers the cell body of the whole cell. The time-lapse was checked to ensure that the correct cells were being measured to verify that the cells did not shift during solution changes. The time measurement software (NIS Software ND Acquisition, NIKON) was utilized to collect time-lapse ratio measurements for each ROI in each image. These measurements were exported to Excel, in which baseline, treatment average, and percent change from baseline calculations were done. The baseline was taken as the average of the F_340_/F_380_ from zero to two minutes when the phosphate buffer was applied. The treatment average calculations were done as the average of the F_340_/F_380_ during each respective treatment application period. The percent response was calculated with the treatment average over the baseline multiplied by 100 to calculate the calcium response change for each treatment.

### 4.10. XTT Viability Assay

XTT Viability Assay was performed using CyQUANT™ XTT Cell Viability Assay kit (ThermoScientific, Waltham, MA, USA Cat# #X12223). Briefly, bEnD.3 cells were cultured in a 96-well plate, at 1 × 10^4^ cells/well seeding density. On the day of the experiment, cells were treated with different concentrations of LEI-106 (100 nM, 1 μM, 10 μM, 100 μM, 325 μM, 625 μM, and 1.3 mM) along with vehicle control for 15 min, followed by XTT assay performed according to the manufacturer’s instructions.

### 4.11. DAG ELISA

DAG ELISA kit (Aviva System Biology, San Deigo, CA, USA, Cat #OKEH02607) was used according to the manufacturer’s instructions. bEnD.3 cells were treated with LEI-106 (1.3 mM) or vehicle for 15 min, followed by a wash step before harvest by cell scraping. The cell pellet was lysed and then centrifuged (5000× *g*, 10 min, 4 °C). The supernatant was applied in the immunoassay.

### 4.12. PKC Kinase Activity Assay

PKC Kinase Activity Assay (ab139437) was used according to the manufacturer’s instructions. bEnd.3 cells were treated with LEI-106 or vehicle for 15 min, as described above. The cells were lysed in lysis buffer and then centrifuged (13,000 rpm, 15 min, 4 °C). The supernatant was subjected to the assay.

### 4.13. Phospho-Proteomics

Mouse bEnd.3 endothelial cells cultured on 150 mm plates were treated with vehicle (0.9% DMSO in media) or LEI-106 (650 μM) for 15 min, then washed with ice-cold PBS. The cells were lysed in ice-cold lysis buffer (20 mM Tris-HCl (pH 7.4), 50 mM NaCl, 2 mM MgCl_2_ hexahydrate, 1% (*v*/*v*) NP40, 0.5% (*w*/*v*) sodium deoxycholate, 0.1% (*w*/*v*) SDS) supplemented with protease and phosphatase inhibitor cocktail (Halt Protease and Phosphatase inhibitor cocktail, ThermoScientific, #78441). The lysate was incubated at 4 °C with continuous rocking for 30 min. After centrifuging at 12,000× *g* for 10 min at 4 °C, the supernatant was collected. Protein concentration was determined using a BCA protein quantitation assay (ThermoScientific). Then, 5 mg of protein lysate per sample (*n* = 4) was subjected to in-solution tryptic digestion and phospho-peptide enrichment using sequential enrichment from metal oxide affinity chromatography per manufacturer’s protocol (Thermo Scientific, San Jose, CA, USA) similar to as previously described to determine global differences in protein phosphorylation abundance between treatments [49]. HPLC-ESI-MS/MS was performed in positive ion mode on a Thermo Scientific Orbitrap Fusion Lumos tribrid mass spectrometer fitted with an EASY-Spray Source (Thermo Scientific) as previously described [50].

Spectra were acquired using XCalibur, version 2.3 (ThermoFisher Scientific). Progenesis QI for proteomics software (version 2.4, Nonlinear Dynamics Ltd., Newcastle upon Tyne, UK) was used to perform ion-intensity-based label-free quantification, as previously described [51,52]. In brief, .raw files were imported and converted into two-dimensional maps (y-axis = time, x-axis = *m*/*z*), followed by the selection of a reference run for alignment purposes. An aggregate data set containing all peak information from all samples was created from the aligned runs, which was then further narrowed down by selecting only +2, +3, and +4 charged ions for further analysis. The samples were then grouped by treatment. A peak list of fragment ion spectra was exported in Mascot generic file (.mgf) format and searched against the mouse SwissProt database using Mascot (Matrix Science, London, UK; version 2.6). The search variables that were used were: 10 ppm mass tolerance for precursor ion masses and 0.5 Da for product ion masses; digestion with trypsin; a maximum of two missed tryptic cleavages; variable modifications of oxidation of methionine and phosphorylation of serine, threonine, and tyrosine; 13C = 1. The resulting Mascot .xml file was then imported into Progenesis, allowing for peptide/protein assignment, while peptides with a Mascot Ion Score of <25 were not considered for further analysis. Precursor ion-abundance values for peptide ions were normalized to all proteins.

Principal component analysis and unbiased hierarchal clustering analysis (heat map) were performed in Perseus [53,54]. Gene ontology and Reactome pathway enrichment analysis was performed with DAVID [55].

### 4.14. Statistical Analysis

GraphPad Prism 7.0 and 8.3.1 software (GraphPad Software) were used for statistical analysis. To determine the numbers needed for each experiment, G.Power3.1 was used for 80% power to detect a 20% difference when alpha = 0.05. The data were expressed as mean ± S.E.M unless otherwise stated. Groups were compared by unpaired *t*-test or one-way ANOVA with Tukey’s post-test, as indicated. Differences were considered significant if *p* ≤ 0.05.

## 5. Conclusions

Clinically, increased permeability of the BBB is associated with worsened prognosis, increased risk of CNS infection, and advanced neuronal loss in neurodegenerative diseases [4]. It is, therefore, imperative to understand the mechanisms by which decreased endocannabinoid tone plays a role in compromising BBB integrity to facilitate novel therapeutic development. Given the above findings, therapies targeting 2-AG hydrolysis or Rho kinases have the potential to meet this need in addition to their ability to treat headache-like pain [15].

## Figures and Tables

**Figure 1 ijms-25-00531-f001:**
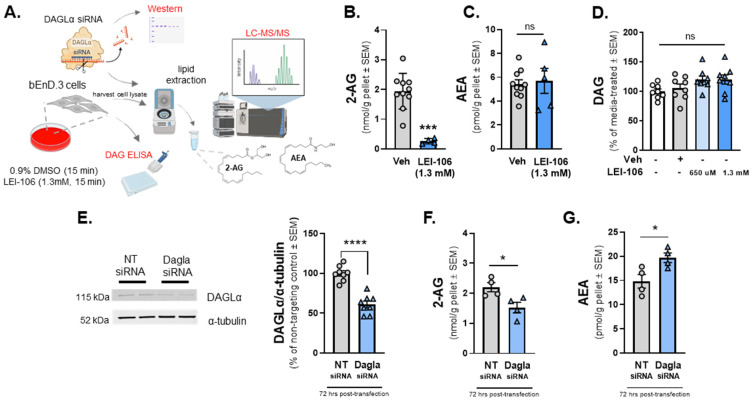
The pharmacological or genetic manipulation of DAGLα decreases the level of 2-AG in bEnD.3 endothelial cells. bEnD.3 endothelial cells were treated with LEI-106 (1.3 mM) or vehicle (0.9% DMSO in media) for 15 min, then subjected to LC-MS/MS to measure 2-AG and AEA levels. In a separate set, the level of DAG was quantified by ELISA. In the siRNA transfection experiments, bEnD.3 cells were transiently transfected with siRNA targeting DAGLα or a scrambled, non-targeting control, then harvested 72 h post-transfection, followed by a Western blot to detect DAGLα expression. In a separate set of experiments, the cell pellet was lysed and subjected to LC-MS to measure 2-AG and AEA levels. (**A**) The schema of the experimental setting. (**B**) The pharmacological blockade of DAGLα with LEI-106 significantly reduced the 2-AG level in endothelial cells (LEI-106 vs. vehicle, *p* = 0.001, t(12) = 5.447 as assessed by unpaired *t*-test). Data are shown as mean ± SEM in nmol/g pellet (*n* = 4-10 in each group). *** denotes significantly different (*p* < 0.001). (**C**) After LEI-106 treatment, the level of AEA was unchanged (LEI-106 vs. vehicle, *p* = 0.7578, t(13) = 0.3149 as assessed by unpaired *t*-test). Data are shown as mean ± SEM in pmol/g pellet (*n* = 5–10 in each group). ns = non-significant. (**D**) The LEI-106 treatment did not significantly change the level of DAG compared to vehicle control (LEI-106, 650 µM vs. vehicle: *p* = 0.4364, t(14) = 0.8011; LEI-106, 1.3 mM vs. vehicle: *p* = 0.4626, t(14) = 0.7553 as assessed by unpaired *t*-test). Data are presented as % of media-treated ± SEM (*n* = 8/group). ns = non-significant. (**E**) Representative image showing the expression of DAGLα with α-tubulin as loading control in bEnd.3 cell lysate harvested 72 h after siRNA transfection. The transfection of DAGLα-targeting siRNA significantly reduced the detection of DAGLα protein compared to the non-targeting control (DAGLα siRNA vs. non-targeting control siRNA: *p* < 0.0001, t(14) = 7.227, as assessed by unpaired *t*-test). Data are shown as % of non-targeting control ± SEM (*n* = 8/group). **** denotes significantly different (*p* < 0.0001). (**F**) siRNA knockdown of DAGLα in bEnD.3 cells significantly reduced 2-AG levels compared to non-targeting control (DAGLα siRNA vs. non-targeting control siRNA: *p* = 0.0266, t(6) = 2.919, as assessed by unpaired *t*-test). Data are shown as mean ± SEM in nmol/g pellet (*n* = 4/group). * denotes significantly different (*p* < 0.05). (**G**) The AEA level of bEnd.3 cells was significantly increased after DAGLα siRNA transfection compared to non-targeting control, suggesting the presence of a compensatory mechanism after the loss of 2-AG (DAGLα siRNA vs. non-targeting control siRNA: *p* = 0.031, t(6) = 2.804, as assessed by unpaired *t*-test). Data are shown as mean ± SEM in pmol/g pellet (*n* = 4/group). * denotes significantly different (*p* < 0.05).

**Figure 2 ijms-25-00531-f002:**
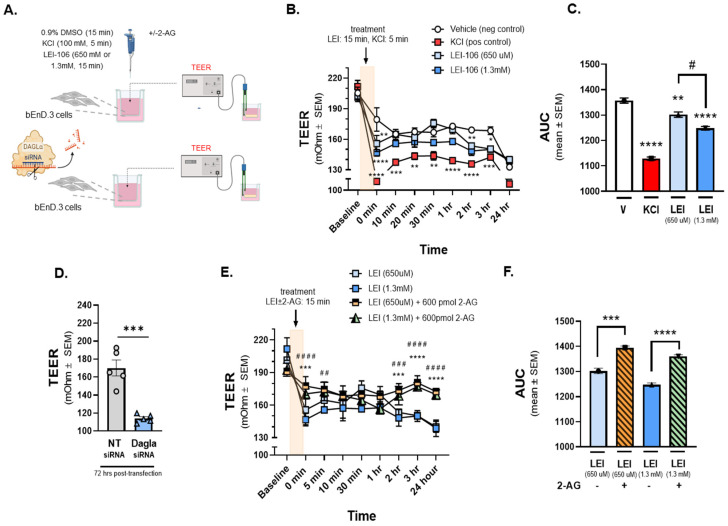
The depletion of 2-AG decreased Trans-Endothelial Electrical Resistance (TEER) of bEnD.3 cells, indicating the loss of barrier integrity. The bEnD.3 endothelial cells were cultured on a trans-well insert, then treated with LEI-106 (650 uM or 1.3 mM) or vehicle (0.9% DMSO in media) with or without 2-AG (600 pmol) for 15 min. KCl pulse (100 mM, 5 min) was used as a positive control. TEER was measured at baseline (before any treatment), right after the treatment (0 min), and then 10, 20, 30 min, 1, 2, 3, and 24 h after post-treatment. In the transfection experiments, TEER was measured 72 h post-transfection. (**A**) This panel shows the schematic experimental setting. (**B**) The application of LEI-106 in either dose significantly decreased the TEER value, compared to vehicle control at 0 min time-point, suggesting the loss of barrier integrity (LEI-106-650 uM vs. vehicle: *p* = 0.0017; LEI-106-1.3 mM vs. vehicle: *p* < 0.0001, as assessed by two-way ANOVA, F(24,72) = 3.723). At later time-points, the TEER was then normalized, followed by an additional significant decrease at 2 and 3 h (2 h time-point: LEI-106-1.3 uM vs. vehicle: *p* = 0.0075; 3 h time-point: LEI-106-650 uM vs. vehicle: *p* = 0.0254 LEI-106-1.3 mM vs vehicle: *p* < 0.022, as assessed by two-way ANOVA, F(24,72) = 3.723). All data represent the mean ± SEM of three independent experiments performed in triplicate. * *p* < 0.05, ** *p* < 0.01, *** *p* < 0.001, **** *p* < 0.0001 compared to vehicle control. (**C**) The area under the curve (AUC) of the corresponding panel B. The AUC analysis further confirmed the significant reduction in TEER induced by LEI-106 treatment (LEI-106-650 uM vs. vehicle: *p* = 0.009; LEI-106-1.3 mM vs. vehicle: *p* < 0.0001; LEI-106-650 uM vs.LEI-106-1.3 mM: *p* = 0.01, as assessed by one-way ANOVA with Tukey post-test, F(3,8) = 125.2). Data are shown mean ± SEM. ** *p* < 0.01, **** *p* < 0.0001 compared to vehicle control. ^#^
*p* < 0.05: LEI-106-650 uM vs. LEI-106-1.3 mM. (**D**) bEnD.3 cells were transfected either with non-targeting control siRNA or siRNA targeting DAGLα. TEER was assessed 72 h post-transfection. A significant decrease in TEER was observed in DAGLα siRNA-transfected cells, compared to non-targeting control, indicating disruption of the barrier integrity after silencing of DAGLα (DAGLα siRNA vs. non-targeting siRNA: *p* = 0.0003, t(8) = 6.203 as assessed by unpaired *t*-test, *n* = 5). Data are shown mean ± SEM. *** *p* < 0.001, compared to the non-targeting control. (**E**) The application of 2-AG diminished the effect of LEI-106 on BBB integrity. Significant differences were observed between LEI-106 vs. LEI-106 + 2-AG groups at 0 min time-point (LEI-106-650 uM vs. LEI-106-650 uM + 2-AG: *p* = 0.0002; hjnnnoLEI-106-1.3 mM vs. LEI-106-1.3 mM + 2-AG: *p* < 0.0001, as assessed by two-way ANOVA, F(24,90) = 7.318). All data represent the mean ± SEM of three independent experiments performed in triplicate. LEI-106-650 uM vs. LEI-106-650 uM + 2-AG: *** *p* < 0.001, **** *p* < 0.0001. LEI-106-1.3 mM vs. LEI-106 1.3 + 2-AG: ^##^
*p* < 0.01, ^###^
*p* < 0.001, ^####^
*p* < 0.0001. (**F**) The area under the curve (AUC) for panel E. The AUC analysis also showed that 2-AG can mitigate the loss of barrier integrity caused by the inhibition of DAGLα (LEI-106-650 uM vs. LEI-106-650 uM + 2-AG: *p* = 0.0001; LEI-106-1.3 mM vs. LEI-106-1.3 mM + 2-AG: *p* < 0.0001, as assessed by one-way ANOVA, F(3,8) = 67.39). Data are shown mean ± SEM. *** *p* < 0.001, **** *p* < 0.0001 compared to the corresponding 2-AG treatment.

**Figure 3 ijms-25-00531-f003:**
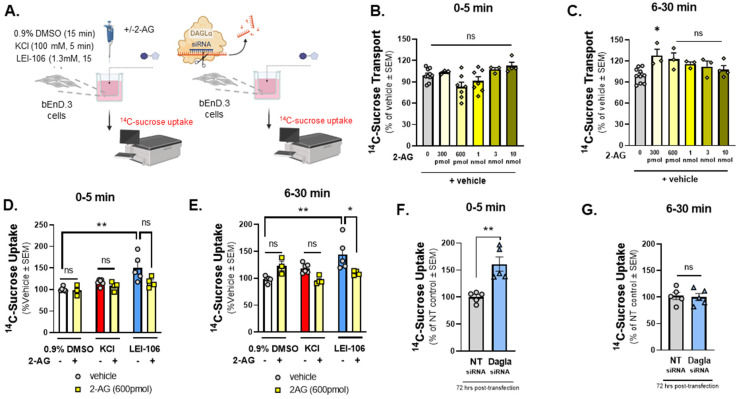
^14^C-Sucrose transport through the bEnd.3 monolayer was increased after the blockade of DAGLα. The bEnD.3 endothelial cells were cultured on a trans-well insert, then treated with an increasing amount of 2-AG (0–10 nmole). In a separate experiment, cells were treated with LEI-106 (1.3 mM) or vehicle (0.9% DMSO in media) with or without 2-AG (600 pmol) for 15 min. KCl pulse (100 mM, 5 min) was used as a positive control. Following treatment, luminal media was replaced with fresh media containing ^14^C-sucrose (0.25 µCi/mL). Abluminal media was collected at 5 min and 30 min post-treatment and subjected to measure radioactivity with a liquid scintillation counter. ^14^C-sucrose uptake assay was performed 72 h after transfection of DAGLα siRNA or non-targeting control. ^14^C-sucrose (0.25 µCi/mL) was added to the luminal side, abluminal media was collected at two time-points (5 min and 30 min), and then radioactivity was measured. (**A**) Schema of experimental setting. (**B**) 2-AG treatment at any dose did not cause a significant change in ^14^C-sucrose uptake at 5 min time-point compared to vehicle control (2-AG at any dose vs. vehicle: *p* > 0.05, F(5,28) = 4.010, as assessed by one-way ANOVA with Bartlett’s test) All data represent the % of vehicle-treated ± SEM (*n* = 3–10/group). ns = non-significant. (**C**) A significant increase in ^14^C-sucrose uptake after 300 pmol of 2-AG treatment was observed at 6–30 min time-point compared to vehicle control (2-AG-300 pmol vs. vehicle: *p* = 0.0172, 2-AG at other doses vs. vehicle: *p* > 0.05, F(5,20) = 3.879, as assessed by one-way ANOVA with Bartlett’s test) All data represent the % of vehicle-treated ± SEM (*n* = 3–10/group). ns = non-significant, * *p* < 0.05. (**D**) LEI-106 (1.3 mM) increased the ^14^C-sucrose uptake at 5 min time-point compared to vehicle control, indicating the presence of paracellular leak (LEI-106 vs. vehicle: *p* = 0.0033, t(9) = 3.965 as assessed by unpaired *t*-test). The application of 2-AG did not significantly reduce the elevated sucrose uptake induced by LEI-106 (LEI-106 vs. LEI-106 + 2-AG: *p* = 0.0797, t(7) = 2.049 as assessed by unpaired *t*-test). All data represent the % of vehicle-treated ± SEM from three-four independent experiments using 4 trans-well inserts/group. ** *p* < 0.01 compared to vehicle control. ns = non-significant. (**E**) Increased ^14^C-sucrose uptake was also observed at the 30 min time-point after LEI-106 treatment (LEI-106 vs. vehicle: *p* = 0.0092, t(8) = 3.415 as assessed by unpaired *t*-test). 2-AG treatment significantly mitigated the increase in sucrose uptake caused by DAGLα inhibition (LEI-106 vs. LEI-106 + 2-AG: *p* = 0.05, t(7) = 2.273 as assessed by unpaired *t*-test). All data represent the % of vehicle-treated ± SEM from three-four independent experiments using 4 trans-well inserts/group. * *p* < 0.05, ** *p* < 0.01 compared to corresponding controls. (**F**) Silencing of DAGLα significantly increased the sucrose uptake in the first 5 min, compared to non-targeting control, suggesting reduced barrier integrity caused by genetic inhibition of DAGLα (DAGLα siRNA vs. non-targeting siRNA: *p* = 0.0022, t(8) = 4.420 as assessed by unpaired *t*-test, *n* = 5). Data are shown mean ± SEM. ** *p* < 0.01, compared to the non-targeting control. (**G**) No significant difference between DAGLα siRNA and control was observed at the later time-point (6–30 min) (DAGLα siRNA vs. non-targeting siRNA: *p* = 0.8204, t(8) = 0.2347 as assessed by unpaired *t*-test, *n* = 5). Data are shown mean ± SEM. ns = non-significant, compared to the non-targeting control.

**Figure 4 ijms-25-00531-f004:**
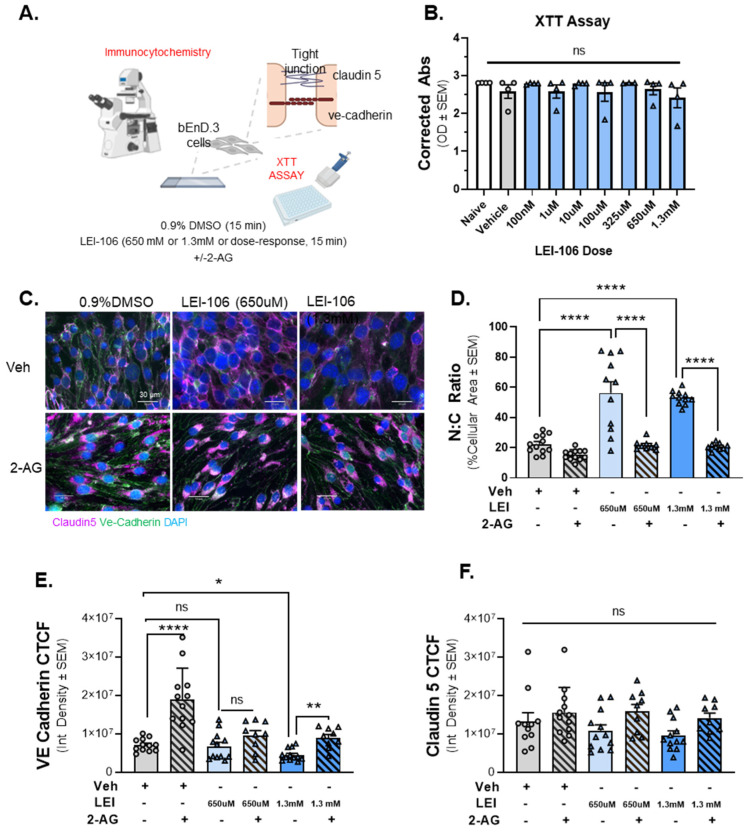
The inhibition of DAGLα induced morphological changes in bEnD.3 cells accompanied with reduced expression of VE-cadherin. Cell viability assay on bEnD.3 cells was performed using increasing doses of LEI-106 (100 nM–1.3 mM). For ICC experiments, the bEnD.3 cells were treated with two doses of LEI-106 (650 uM and 1.3 mM) for 15 min, followed by 2-AG or vehicle application. ICC was performed to detect possible changes in the expression of two tight junction proteins, claudin 5 and VE-cadherin. (**A**) The panel represents the setting of the experiments. (**B**) LEI-106 did not cause significant changes in the cell viability at any doses applied in the study. (LEI-106 at any doses vs. vehicle: *p* > 0.05, as assessed by one-way ANOVA with Bartlett post-test, F(8,26) = 0.7738). All data are expressed as the mean of optical density ± SEM (*n* = 4 in each group). ns = non-significant. (**C**) Representative ICC images of bEnd.3 cells treated with LEI-106 ±2-AG application. (**D**) Significant changes in the nuclear: cytoplasmic ratio were observed following LEI-106 treatment, suggesting possible morphological alteration caused by 2-AG depletion (LEI-106-650 µM vs. vehicle: *p* < 0.0001, LEI-106-1.3 mM vs. vehicle: *p* < 0.0001, as assessed by one-way ANOVA with Tukey post-test, F(8,85) = 30.50). 2-AG treatment following LEI-106 application significantly reversed these effects (LEI-106-650 µM vs. LEI-106-650 µM + 2-AG: *p* < 0.0001, LEI-106-1.3 mM vs. LEI-106-1.3 mM + 2-AG: *p* < 0.0001, as assessed by one-way ANOVA with Tukey post-test, F(8,85) = 30.50). All data are shown as the mean % of cellular area ± SEM (*n* = 9–12/condition). **** *p* < 0.0001, compared to corresponding controls. (**E**) The higher dose of LEI-106 (1.3 mM) significantly reduced VE-cadherin CTCF; however, no significant change was detected at the lower dose (650 µM) compared to vehicle control (LEI-106-650 µM vs. vehicle: *p* = 0.9052, LEI-106-1.3 mM vs. vehicle: *p* = 0.0233, as assessed by one-way ANOVA with Bartlett post-test, F(2,33) = 4.100). The administration of 2-AG increased the VE-cadherin CTCF, moderating the loss of VE-cadherin expression after LEI-106 treatment (LEI-106-1.3 mM vs. LEI-106-1.3 mM + 2-AG: *p* = 0.005, as assessed by one-way ANOVA with Tukey post-test, F(4,49) = 5.568). All values are expressed as the mean of corrected total cell fluorescence (CTCF) ± SEM (*n* = 9–12/condition). * *p* < 0.05, ** *p* < 0.01, **** *p* < 0.0001, compared to the corresponding controls. ns = non-significant. (**F**) Neither dose of LEI-106 treatment significantly influenced the CTCF of claudin-5 compared to vehicle control (LEI-106-650 µM vs. vehicle: *p* = 0.8982, LEI-106-1.3 mM vs. vehicle: *p* = 0.6161, as assessed by one-way ANOVA with Tukey post-test, F(5,59) = 2.276). All values are expressed as the mean of corrected total cell fluorescence (CTCF) ± SEM (*n* = 9–12/condition). ns = non-significant.

**Figure 5 ijms-25-00531-f005:**
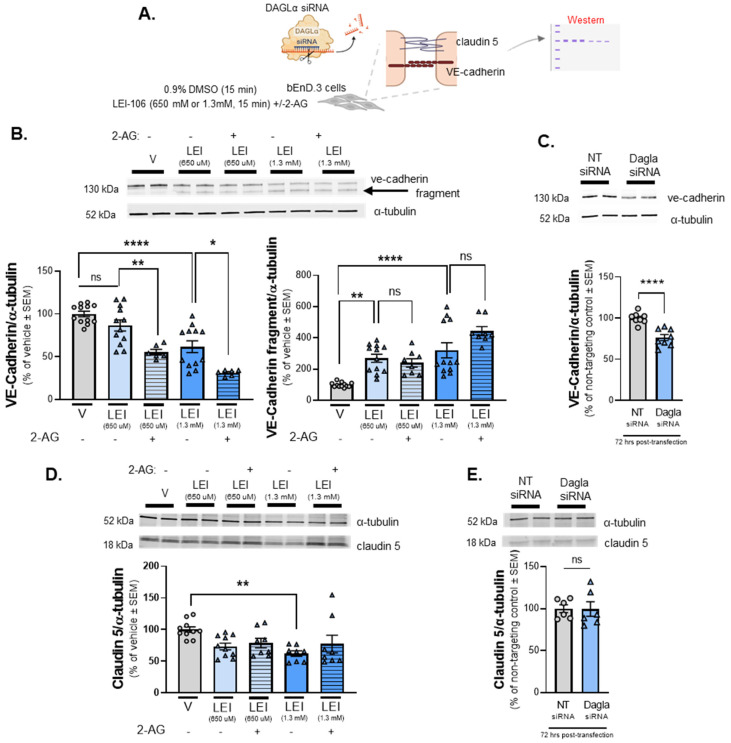
The detection of VE-cadherin was altered after DAGLα blockade in bEnd.3 cells. The bEnD.3 cells were treated with LEI-106 (650 µM or 1.3 mM) for 15 min with or without 2-AG (600 pmol) application. The cell lysate was subjected to Western to detect the expression of VE-cadherin and claudin 5. In the siRNA transfection experiment, samples were harvested 72 h post-transfection and subjected to Western to detect the expression of VE-cadherin and claudin 5. (**A**) Scheme of experimental outline. (**B**) Representative image showing the expression of VE-cadherin with α-tubulin used as loading control. The pharmacological blockade of DAGLα induced fragmentation of VE-cadherin. The treatment of LEI-106 at 650 µM dose did not cause significant change in the detection of VE-cadherin main band compared to vehicle control (vehicle vs. LEI-106, 650 µM: *p* = 0.3565 as assessed by one-way ANOVA with Bartlett’s test, F(4,43) = 19.62), but it significantly increased the detection of VE-cadherin fragment (vehicle vs. LEI-106, 650 µM: *p* = 0.0014 as assessed by one-way ANOVA with Bartlett’s test, F(4,47) = 15.19). The higher dose of LEI-106 (1.3 mM) significantly reduced the detection of VE-cadherin main band and increased the detection of the fragmented one compared to vehicle control (main band: vehicle vs. LEI-106-1.3 mM: *p* < 0.0001 as assessed by one-way ANOVA with Bartlett’s test, F(4,43) = 19.62; fragment: vehicle vs. LEI-106, 1.3 mM: *p* < 0.0001 as assessed by one-way ANOVA with Bartlett’s test, F(4,47) = 15.19). The application of 2-AG further decreased the detection of main VE-cadherin band, but it did not significantly influence the fragmentation compared to corresponding controls (main band: LEI-106, 650 µM vs. LEI-106, 650 µM + 2-AG: *p* = 0.0092, LEI-106, 1.3 mM vs. LEI-106, 1.3 mM + 2-AG: *p* = 0.011 as assessed by one-way ANOVA with Bartlett’s test, F(4,43) = 19.62; fragment: LEI-106-650 µM vs. LEI-106, 650 µM + 2-AG: *p* = 0.9665, LEI-106-1.3 mM vs. LEI-106-1.3 mM + 2-AG: *p* = 0.0769 as assessed by one-way ANOVA with Bartlett’s test, F(4,47) = 15.19). All data presented as % of vehicle-treated ± SEM (*n* = 8–12/condition). * *p* < 0.05, ** *p* < 0.01, **** *p* < 0.0001, compared to the corresponding controls. ns = non-significant. (**C**) Representative image showing the expression of VE-cadherin with α-tubulin as loading control in transfected cell samples. The transfection of DAGLα-specific siRNA significantly reduced the detection of VE-cadherin compared to the non-targeting control (DAGLα siRNA vs. non-targeting control siRNA: *p* < 0.0001, t(14) = 5.376, as assessed by unpaired *t*-test). Data are shown as % of non-targeting control ± SEM (*n* = 8/group). **** denotes significantly different (*p* < 0.0001). (**D**) Representative image displaying the expression level of claudin 5. α-tubulin was used as a loading control. The lower dose of LEI-106 (650 µM) did not significantly influence the detection of claudin 5 compared to vehicle control (vehicle vs. LEI-106-650 µM: *p* = 0.0759 as assessed by one-way ANOVA with Tukey’s test, F(4,39) = 3.559). The treatment of LEI-106 at 1.3 mM dose significantly reduced the detection of claudin 5 compared to vehicle control (vehicle vs. LEI-106-1.3 mM: *p* = 0.0076 as assessed by one-way ANOVA with Tukey’s test, F(4,39) = 3.559). The application of 2-AG did not cause significant changes compared to the corresponding controls (LEI-106-650 µM vs. LEI-106-650 µM + 2-AG: *p* = 0.9870, LEI-106-1.3 mM vs. LEI-106-1.3 mM + 2-AG: *p* = 0.6535 as assessed by one-way ANOVA with Tukey’s test, F(4,39) = 3.559). All data presented as % of vehicle-treated ± SEM (*n* = 8-10/condition). ** *p* < 0.01, compared to vehicle control. (**E**) Representative image of the expression of claudin 5 along with α-tubulin as loading control in transfected cell samples. No significant difference in the detection of claudin 5 was observed after the transfection of DAGLα-specific siRNA (DAGLα siRNA vs. non-targeting control siRNA: *p* = 0.9666, t(10) = 0.04292, as assessed by unpaired *t*-test). Data are shown as % of non-targeting control ± SEM (*n* = 6/group). ns = non-significant.

**Figure 6 ijms-25-00531-f006:**
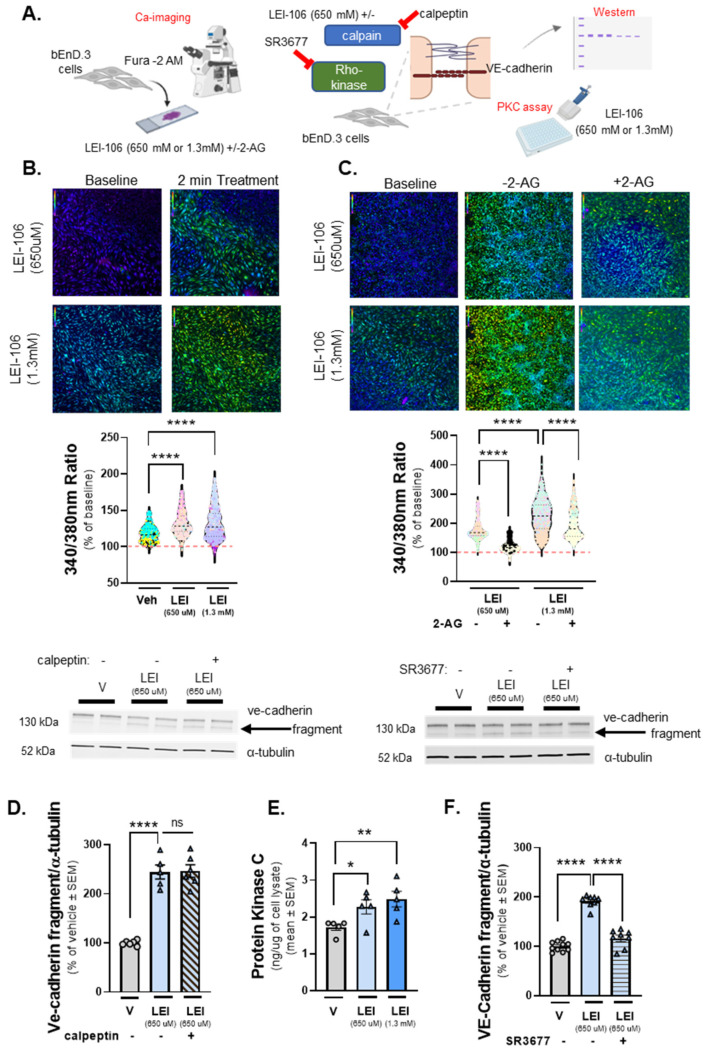
Increased intracellular calcium level and elevated PKC activity after DAGLα inhibition in bEnD.3 cells. The bEnd.3 cells plated on collagen-coated coverslips were subjected to calcium imaging with the dye Fura-2. After 2-min baseline observation in buffer, cells were treated with LEI-106 (650 µM or 1.3 mM) or vehicle (0.9% DMSO) for 2 min, followed by a subsequent 2-min washout phase. In a separate experiment, cells received buffer spiked with 2-AG (600 pmol/coverslip) during the washout phase. Images used for analysis were captured at the beginning of the treatment application and after removal of the treatment, approximately 2 min apart. In a separate experiment, protein kinase C (PKC) activity was measured by ELISA in bEnD.3 cells after LEI-106 treatment. bEnD.3 cells were treated with calpain inhibitor, calpeptin (10 µM) or Rho-kinase inhibitor, SR3677 (100 nM) 30 min prior to the application of LEI-106 (650 µM), then subjected to Western blot to detect changes in the fragmentation of VE-cadherin caused by DAGLα inhibition. (**A**) Schema of experimental setting. (**B**) Representative calcium images of bEnD.3 cells at baseline, then treated with two different doses of LEI-106 (650 µM, 1.3 mM). LEI-106 at both doses significantly increased the intracellular calcium levels compared to vehicle control (LEI-106-650 µM vs. vehicle: *p* < 0.0001, LEI-106-1.3 mM vs. vehicle: *p* < 0.0001, as assessed by one-way ANOVA with Tukey post-test, F(3,3295) = 482.1). All values are % of baseline obtained from three independent experiments using three coverslips in each. In each coverslip, 100 cells were analyzed by FIJI Image J software 1.54. **** *p* < 0.0001 compared to vehicle control. Red dashed line represents the baseline. (**C**) Representative images showing intracellular calcium levels in bEnD.3 cells treated with LEI-106 (650 µM or 1.3 mM) with or without 2-AG. The application of 2-AG significantly reduced the elevation of intracellular calcium caused by LEI-106 treatment (LEI-106-650 µM vs. LEI-106-650 µM + 2-AG: *p* < 0.0001, LEI-106-1.3 mM vs. LEI-106-1.3 mM + 2-AG: *p* < 0.0001, as assessed by one-way ANOVA with Tukey post-test, F(3,2700) = 889.9). All data are shown as % of baseline obtained from three independent experiments using three coverslips in each. In each coverslip, 100 cells were analyzed by Image J software. **** *p* < 0.0001 compared to vehicle control. Red dashed line represents the baseline. (**D**) Representative image of Western blot showing VE-cadherin expression in bEnD.3 cells treated with LEI-106 with or without calpain inhibitor, calpeptin (10 µM). α-tubulin was used as a loading control. The calpain inhibitor did not significantly change the fragmentation of VE-cadherin caused by LEI-106 treatment (LEI-106-650 µM vs. LEI-106-650 µM + calpeptin: *p* > 0.9999, as assessed by one-way ANOVA with Tukey post-test, F(4,20) = 84.40). All data are shown as % of vehicle-treated ± SEM (*n* = 5–6/condition). **** *p* < 0.0001, ns = non-significant. (**E**) The assessment of PKC activity by ELISA showed that LEI-106 treatment (15 min) significantly increased the activity level of PKC compared to vehicle control (LEI-106-650 µM vs. vehicle: * *p* = 0.0289, t(8) = 2.658; LEI-106-1.3 mM vs. vehicle: ** *p* = 0.0099, t(8) = 3.362 as assessed by unpaired *t*-test). Data are presented as the mean of PKC in ng/µg of cell lysate ± SEM (*n* = 5/group). (**F**) Representative image of immunoblot targeting VE-cadherin and α-tubulin as loading control in bEnD.3 cells treated with LEI-106 (650 µM) with or without Rho-kinase inhibitor, SR3677 (100 nM). The pretreatment (30 min) with Rho-kinase inhibitor significantly mitigated the fragmentation of VE-cadherin caused by LEI-106 (LEI-106-650 µM vs. LEI-106-650 µM + SR3677: *p* < 0.0001, as assessed by one-way ANOVA with Tukey post-test, F(2,23) = 110.3). All data are shown as % of vehicle-treated ± SEM (*n* = 8–10/condition). **** *p* < 0.0001, compared to the LEI-106 (650 µM) treatment.

**Figure 7 ijms-25-00531-f007:**
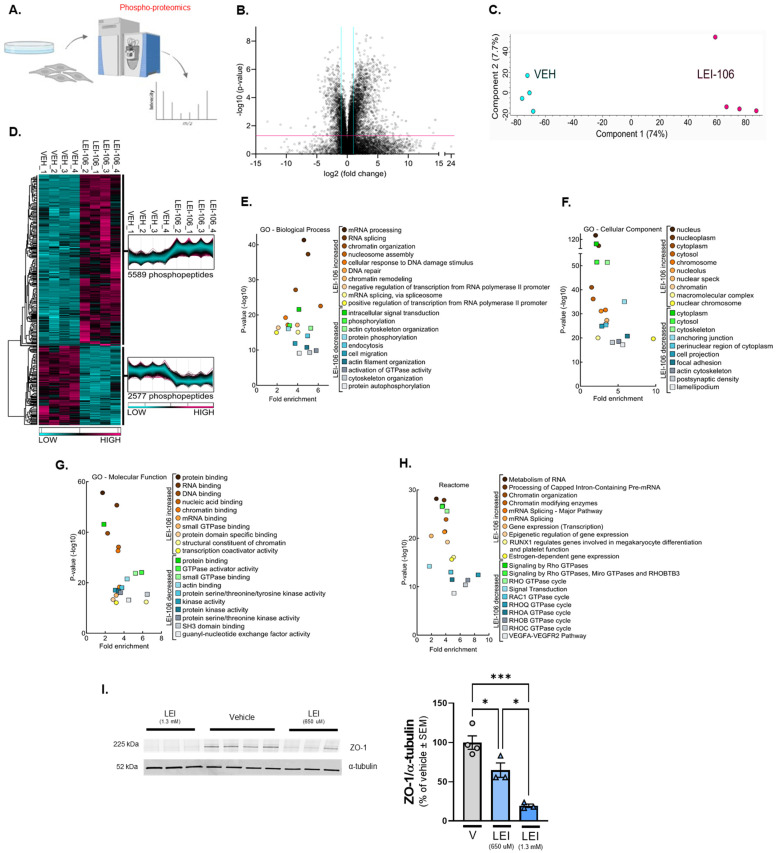
Quantitative phospho-proteomics after DAGLα inhibition in bEnD.3 cells. bEnd.3 cells cultured on 150 mm plates were treated with vehicle (0.9% DMSO in media) or LEI-106 (650 µM) for 15 min, then subjected to quantitative phospho-proteomics. (**A**) Schema of the experiment. (**B**) Volcano plot of bEnd.3 of detected phospho-peptides showing a greater than 2-fold change after LEI-106 compared to vehicle treatment. (**C**) Principal Component Analysis (PCA) of cells treated with vehicle or LEI106 (650 µM). (**D**) Heatmap of phospho-peptides increased by LEI-106 treatment compared to vehicle or decreased compared to vehicle. (**E**) GO Biological process functional analysis showing the top 10 processes altered in each condition. (**F**) GO cellular compartment functional analysis showing the top 10 compartments in which phospho-peptides were impacted in each condition. (**G**) GO molecular function functional analysis shows the top 10 in which phospho-peptides were impacted in each condition. (**H**) Top 10 Reactome Pathways associated with changes in phospho-peptides in each outcome. *n* = 4 biological replicates per condition. (**I**) bEnd.3 cells were treated with LEI-106 (650 µM or 1.03 mM) or vehicle for 15 min, then subjected to Western to detect ZO-1. Representative image showing the expression of ZO-1 along with α-tubulin as a loading control. The blockade of DAGLα significantly decreased the detection of ZO-1, compared to vehicle control (vehicle vs. LEI-106-650 µM: *p* = 0.034, vehicle vs. LEI-106-1.3 mM: *p* = 0.0004, LEI-106-650 µM vs. LEI-106-1.3 mM: *p* = 0.0142 as assessed by one-way ANOVA with Tukey post-test, F(2,7) = 27.41). All data presented as % of vehicle-treated ± SEM (*n* = 3–4/condition). * *p* < 0.05, *** *p* < 0.001.

**Figure 8 ijms-25-00531-f008:**
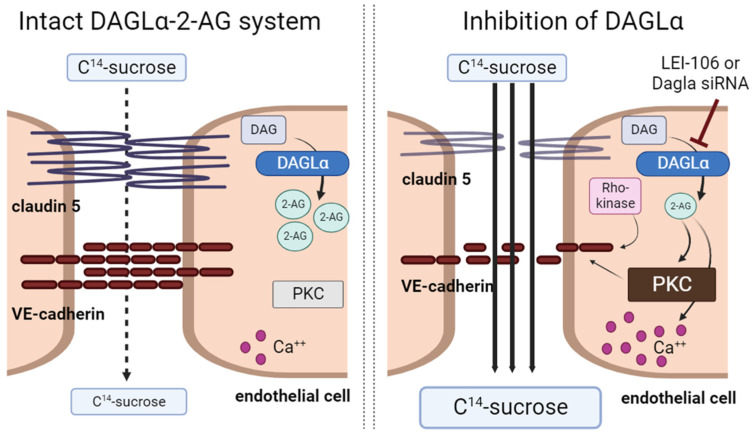
2-AG depletion via inhibition of DAGLα disrupted integrity of bEnd.3 endothelial cells. The inhibition of DAGLα either with LEI-106 or siRNA led to loss of 2-AG and induced paracellular leak manifesting by increased C^14^-sucrose transport through the monolayer of bEnd.3 cells. Disruption of 2-AG homeostasis was connected to increased intracellular calcium signaling, enhanced activity of protein kinase C (PKC), and fragmentation of tight junction protein, VE-cadherin. Inhibition of Rho-kinase mitigated the fragmentation of VE-cadherin induced by the blockade of DAGLα.

## Data Availability

The datasets generated and analyzed during the current study are available from the corresponding author upon reasonable request.

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
