# Peer review of "Depletion of Endothelial-Derived 2-AG Reduces Blood-Endothelial Barrier Integrity via Alteration of VE-Cadherin and the Phospho-Proteome"

_ijms, 2023, doi:10.3390/ijms25010531_

Round 1

Reviewer 1 Report

Comments and Suggestions for Authors

This study describes roles for 2-AG in controlling blood-brain barrier function and presents mechanistic evidence for V-cadherin in dysfunction occurring after pharmacological and genetic reduction of activity off the 2-AG biosynthetic enzyme, DGL. The experiments are generally well conducted and the results match the conclusions. I only have a few recommendations to enhance the presentation of some data and inclusion of additional discussion points. 

1) Levels of 2-AG and AEA were quantitate via LC/MS using adequate internal standards and associated isotope dilution method, as described in the methods. This is good and is a reliable method routinely used in the field. My concern is that the data is presented as % of vehicle. Please revise the graphs to include actual amounts (i.e., pmol per g or nmol per g). This shouldn't be an issue given that the experiments were conducted with care using the appropriate quantitative methods. 

2) While the study used cannabinoid receptor agonists to identify roles for CB1 and/or CB2 in the effects, it would be ideal to combine use of the agonists with selective CB1 and CB2R antagonists to identify roles for specific receptors in these effects. I appreciate the effort it takes for these experiments so my only request here is to add a discussion point about this being a limitation to identifying specific receptors, and pose these experiments for the future. 

3) There have been several studies identifying roles for the endocannabinioid system in controlling gut-barrier function. as well. Please add a small discussion comparing and contrasting possible roles for the system in maintaining blood-barrier integrity as well as gut barrier integrity. In fact, a recent paper show that rodent obesity is associated with lower levels of 2-AG and DGL activity in the large intestine, which may underlie compromised gut-barrier function found in obesity (PMID: 36142461). Roles for CB1 were also described.

Author Response

We thank the reviewer for their insightful comments. We have responded in bold below to each point.

1) Levels of 2-AG and AEA were quantitate via LC/MS using adequate internal standards and associated isotope dilution method, as described in the methods. This is good and is a reliable method routinely used in the field. My concern is that the data is presented as % of vehicle. Please revise the graphs to include actual amounts (i.e., pmol per g or nmol per g). This shouldn't be an issue given that the experiments were conducted with care using the appropriate quantitative methods. – In the revised version, the actual amount of endocannabinoid levels (nmol/g pellet for 2-AG and pmol/g pellet for AEA) are presented. We also noticed that we reported wrong numbers for basal endocannabinoid levels. We do apologize for that mistake and correct numbers are reported in the revised version.

2) While the study used cannabinoid receptor agonists to identify roles for CB1 and/or CB2 in the effects, it would be ideal to combine use of the agonists with selective CB1 and CB2R antagonists to identify roles for specific receptors in these effects. I appreciate the effort it takes for these experiments so my only request here is to add a discussion point about this being a limitation to identifying specific receptors, and pose these experiments for the future.- The role of cannabinoid receptors was discussed in the “Limitation” section of the revised manuscript. (line 720)

3) There have been several studies identifying roles for the endocannabinioid system in controlling gut-barrier function. as well. Please add a small discussion comparing and contrasting possible roles for the system in maintaining blood-barrier integrity as well as gut barrier integrity. In fact, a recent paper show that rodent obesity is associated with lower levels of 2-AG and DGL activity in the large intestine, which may underlie compromised gut-barrier function found in obesity (PMID: 36142461). Roles for CB1 were also described.- A paragraph discussing the role of ECB system in gut barrier function was added in the revised version of manuscript. (lines 704-714).

Reviewer 2 Report

Comments and Suggestions for Authors

In the manuscript by Levine et al., the authors discussed the role of endothelial-derived 2-AG in maintaining the homeostasis of BBB endothelial cells by regulating cell structure and VE-cadherin function.  The data presented in the manuscript is solid and interesting.   The manuscript is very nicely written.  I have a few major/minor suggestions for polishing the manuscript.

Major Comments:

1.     In Figure 7, the authors mentioned that ‘LEI-106 significantly increased phosphorylation proteins involved in mRNA processing and binding as well as pathways regulating mRNA splicing, RUNX1 mediated gene regulation, and estrogen dependent gene expression’.  Did they re-validate these targets through western blot or any other experiments.  The data is very valuable as it can further add up to the novelty of the work.

2.     The authors should provide a more comprehensive explanation of the results presented in Figure 7, as these findings will be crucial for guiding future research works.

Minor Comments:

1.     The authors should avoid repeating the data, (such as the comparison between LEI-106-650 uM and the vehicle (p=0.0017) and LEI-106-1.3 mM and vehicle (p<0.0001), as they have already been mentioned in the figure legends.  The same thing should be avoided in the entire result section.

2.     Figure 2B appears to be somewhat crowded. The authors should consider replacing the figure.

3.     In Figure 3B and 3C, what do the authors mean when they refer to "0-5 minutes" and "6-30 minutes"?  Is it 5 minutes or 30 minutes?

4.     The scale bar in Figure 4C (2-AG, 0.9%DMSO) is missing.

5.     The authors should explain the importance of Zonula Occludens proteins before introducing them in the results.  Also, they should include Supplementary Figure 2 in the main figure.

Author Response

We thank the reviewer for their constructive comments. Our answers to each point can be found in bold below.

Major Comments:

  1. In Figure 7, the authors mentioned that ‘LEI-106 significantly increased phosphorylation proteins involved in mRNA processing and binding as well as pathways regulating mRNA splicing, RUNX1 mediated gene regulation, and estrogen dependent gene expression’.Did they re-validate these targets through western blot or any other experiments.  The data is very valuable as it can further add up to the novelty of the work.- We do agree with the reviewer about the importance of re-evaluation of proteomics data using other techniques, however conducting those experiments were beyond the scope of the current manuscript. We are planning to incorporate those results into our next paper, as it is indicated in the discussion section.
  2. The authors should provide a more comprehensive explanation of the results presented in Figure 7, as these findings will be crucial for guiding future research works.- We thank the reviewer for their suggestion.  The explanation of results for Figure 7 has been expanded in the appropriate section.

Minor Comments:

  1. The authors should avoid repeating the data, (such as the comparison between LEI-106-650 uM and the vehicle (p=0.0017) and LEI-106-1.3 mM and vehicle (p<0.0001), as they have already been mentioned in the figure legends.  The same thing should be avoided in the entire result section. – The journal require us to provide precise description of experimental data in the Results section, which includes reporting of statistical numbers. In the Result section, we reported only p and n numbers. However, in the Figure legends, all statistical numbers were presented. Since we used two different concentrations of LEI-106, reporting statistical data for each is necessary and it should not be considered repeat of data.
  2. Figure 2B appears to be somewhat crowded. The authors should consider replacing the figure. - In the revised version, modification on the scales of Fig. 2B and 2E were made to increase the visibility of individual data points.
  3. In Figure 3B and 3C, what do the authors mean when they refer to "0-5 minutes" and "6-30 minutes"?  Is it 5 minutes or 30 minutes? – The samples were collected at 5 min and at 30 min, however we used 0-5 min and 6-30 min, because the 5 min collection represents the sucrose transport in the first 5 min (0-5 min) and the 30 min collection represents the sucrose transport happening between 6-30 min. We do apologize for the lack of this clarification. We added this detail in the Method section (line 833).
  4. The scale bar in Figure 4C (2-AG, 0.9%DMSO) is missing. – The scale bar was added in the revised figure.
  5. The authors should explain the importance of Zonula Occludens proteins before introducing them in the results.  Also, they should include Supplementary Figure 2 in the main figure. – An introduction about ZO-1 protein was included in the revised version (line 585) and Suppl Fig2 was added to Fig 7.

Round 2

Reviewer 2 Report

Comments and Suggestions for Authors

In the manuscript by Levine et al., the authors discussed the role of endothelial-derived 2-AG in maintaining the homeostasis of BBB endothelial cells by regulating cell structure and VE-cadherin function. 

The authors have addressed all the previous comments. Thus, the manuscript can be accepted in its present form.